# Nucleophosmin integrates within the nucleolus via multi-modal interactions with proteins displaying R-rich linear motifs and rRNA

Diana M Mitrea[1], Jaclyn A Cika[1,2], Clifford S Guy[3], David Ban[1†], Priya R Banerjee[4], Christopher B Stanley[5], Amanda Nourse[1,6], Ashok A Deniz[4], Richard W Kriwacki[1,7]*

[1]Department of Structural Biology, St. Jude Children's Research Hospital, Memphis, United States; [2]Integrative Biomedical Sciences Program, University of Tennessee Health Sciences Center, Memphis, United States; [3]Department of Immunology, St. Jude Children's Research Hospital, Memphis, United States; [4]Department of Integrative Structural and Computational Biology, The Scripps Research Institute, La Jolla, United States; [5]Biology and Biomedical Sciences Group, Biology and Soft Matter Division, Oak Ridge National Laboratory, Oak Ridge, United States; [6]Molecular Interactions Analysis Shared Resource, St. Jude Children's Research Hospital, Memphis, United States; [7]Department of Microbiology, Immunology and Biochemistry, University of Tennessee Health Sciences Center, Memphis, United States

**\*For correspondence:** richard.kriwacki@stjude.org

**Present address:** [†]Cancer Center, University of Louisville, Louisville, United States

**Competing interests:** The authors declare that no competing interests exist.

**Abstract** The nucleolus is a membrane-less organelle formed through liquid-liquid phase separation of its components from the surrounding nucleoplasm. Here, we show that nucleophosmin (NPM1) integrates within the nucleolus via a multi-modal mechanism involving multivalent interactions with proteins containing arginine-rich linear motifs (R-motifs) and ribosomal RNA (rRNA). Importantly, these R-motifs are found in canonical nucleolar localization signals. Based on a novel combination of biophysical approaches, we propose a model for the molecular organization within liquid-like droplets formed by the N-terminal domain of NPM1 and R-motif peptides, thus providing insights into the structural organization of the nucleolus. We identify multivalency of acidic tracts and folded nucleic acid binding domains, mediated by N-terminal domain oligomerization, as structural features required for phase separation of NPM1 with other nucleolar components in vitro and for localization within mammalian nucleoli. We propose that one mechanism of nucleolar localization involves phase separation of proteins within the nucleolus.

## Introduction

The nucleolus, a membrane-less organelle, is the site of ribosome biogenesis and a cellular stress sensor (*Boisvert et al., 2007*). Nucleoli contain three substructures: the fibrillar centers (FCs) and dense fibrillar component (DFC) are engulfed in the granular component (GC) (*Boisvert et al., 2007*), which exhibits ATP-dependent liquid-like features (*Brangwynne et al., 2011*). Ribosomal RNA (rRNA) genes are transcribed between the FC and DFC, and rRNAs are processed while migrating into the GC, wherein they assemble with ribosomal proteins to form pre-ribosomal particles (*Boisvert et al., 2007*). Nucleophosmin (NPM1, also known as B23), a highly abundant marker of the

**eLife digest** Inside cells, machines called ribosomes assemble proteins from building blocks known as amino acids. Cells can alter the numbers of ribosomes they produce to match the cell's demand for new proteins. For instance, when cells grow they require a lot of new proteins and therefore more ribosomes are produced. However, when cells face harsh conditions that cause stress (e.g. exposure to UV radiation or a harmful chemical) they generally stop growing and therefore need fewer ribosomes.

In human and other eukaryotic cells, ribosomes are assembled in a structure called the nucleolus. However, because the nucleolus is not separated from the rest of the cell by a membrane, it was not clear how it is able to accumulate large quantities of the proteins and other molecules needed to make ribosomes. Recent work suggests that the nucleolus is formed through a process referred to as "phase separation" in which the liquid in a particular region of the cell has different physical properties to the liquid surrounding it. This is like how oil and water form separate layers when mixed.

A protein called nucleophosmin is found at high levels in the nucleolus where it interacts with many other proteins, including those involved in making ribosomes. Nucleophosmin binds to motifs within these proteins that contain multiple copies of an amino acid called arginine (referred to as R-motifs). Now, Mitrea et al. investigate how nucleophosmin binds to R-motif proteins and whether this is important for assembling the nucleolus. A search for R-motifs in a list of over a hundred proteins known to bind to nucleophosmin showed that the majority of these proteins contained multiple R-motifs. Furthermore, when high levels of nucleophosmin and the R-motif proteins were present, they underwent phase separation.

Next, Mitrea et al. examine the changes in how nucleophosmin and a ribosomal protein interact before and after phase separation. The experiments show that many molecules of nucleophosmin bind to each other and that multiple regions in nucleophosmin are able to interact with the R-motifs. Together, these interactions produce large assemblies of proteins that result in the creation of separate liquid layers. Furthermore, the experiments show that R-motif proteins and other molecules needed to make ribosomes can be brought together within the same liquid phase by nucleophosmin. Mitrea et al.'s findings provide the first insights into the role of nucleophosmin in the molecular organisation of the nucleolus. The next challenge is to understand how this organisation promotes the production of ribosomes and helps the cell to respond to stressful situations.

GC, functions as a nucleolar chaperone and plays a role in cellular stress responses (*Colombo et al., 2011*). While appreciated (*Brangwynne et al., 2011*; *Chen and Huang, 2001*; *Negi and Olson, 2006*; *Weber and Brangwynne, 2015*), the molecular basis of the GC's fluidity is unknown.

While *Npm1* loss is embryonic lethal in mice, mouse fibroblasts derived from $Npm1^{-/-}/Trp53^{-/-}$ embryos readily proliferate in culture (*Colombo et al., 2005*), indicating that NPM1 is dispensable for ribosome biogenesis. However, NPM1 is known to influence ribosome biogenesis, genome stability and tumor suppression (*Lindstrom, 2011*) and to participate in responses to cellular stresses, including DNA damage (*Lee et al., 2005*), chemotoxicity (*Chan, 1992*; *Yao et al., 2010b*), and oxidative stress (*Paron et al., 2004*). Furthermore, NPM1 depletion is associated with disruption of nucleolar structure (*Holmberg Olausson et al., 2014*). We propose that NPM1 participates in the organization of the liquid-like structure of the GC and consequently may actively participate in stress signal integration and transmission, thereby explaining its known roles in ribosome biogenesis, tumor suppression and other processes (*Lindstrom, 2011*). Accordingly, NPM1 interacts with a vast array of partners (http://thebiogrid.org/110929/summary/homo-sapiens/npm1.html), many involved in ribosome biogenesis, and controls the nucleolar localization of ribosomal (*Lindstrom, 2012*; *Rosorius et al., 2000*), viral (*Duan et al., 2014*; *Fankhauser et al., 1991*), and certain tumor suppressor proteins (*Bertwistle et al., 2004*), many of which engage the N-terminal oligomerization domain (OD) of NPM1 (*Figure 1a*) via arginine-rich short linear motifs (R-motifs) (*Mitrea et al., 2014*). Furthermore, NPM1 binds nucleic acids (e.g., rRNA and DNA) through its C-terminal domain

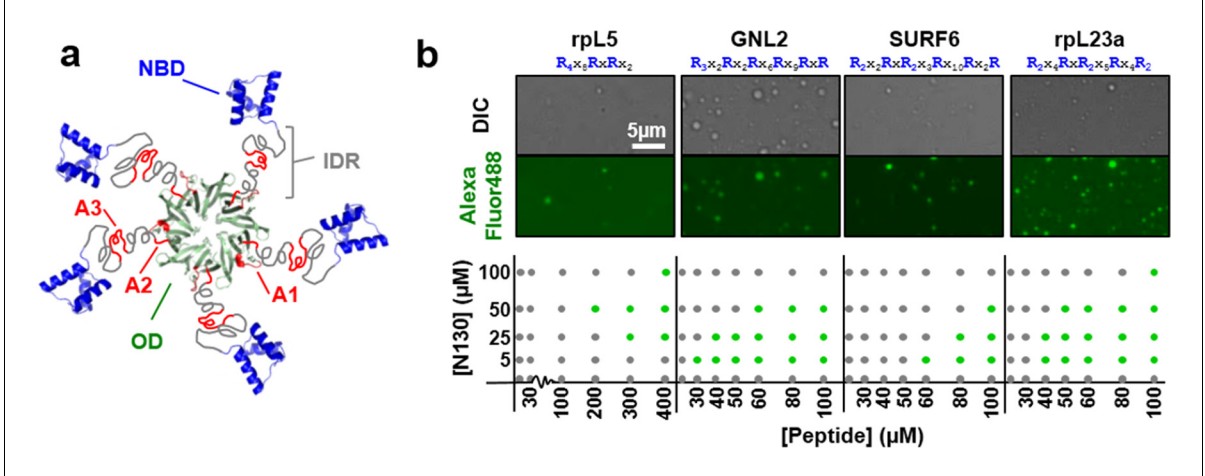

**Figure 1.** Multivalency of acidic tracts within NPM1 and R-motifs within nucleolar substrates mediates liquid-liquid phase separation. (**a**) Composite model of NPM1 structure; the oligomerization domain (OD, green, PDB ID 4N8M), containing the A1 acidic tract (red), is connected via a disordered region (IDR, grey), containing two additional acidic tracts (A2 & A3, red), to the C-terminal nucleic acid binding domain (NBD, blue, PDB ID 2VXD); (**b**) Phase separation diagrams for mixtures of N130 and four R-motif containing peptides (bottom dot graphs); phase separation was assessed by the formation of liquid-like droplets detected using light microscopy (grey dot, clear solution observed; green dot, liquid-like droplets observed). Representative examples of liquid-like droplets formed between 50 µM N130 and the lowest peptide concentration associated with phase separation, visualized by DIC (top panel) and Alexa Fluor488 emission of labeled N130 (bottom panel) are illustrated. The composition of the R-motif peptides is given at the top of each pair of images; R is arginine; and X is any other amino acid.

The following figure supplements are available for figure 1:

**Figure supplement 1.** Representative ITC curves for titrations of multivalent R-motif containing peptides into N122 and N130.

**Figure supplement 2.** Multivalent R-motifs reside in disordered regions of proteins.

(nucleic acid binding domain, NBD) (*Wang et al., 1994*). We seek to understand the molecular mechanisms of NPM1's multifarious functional interactions. Here, we show that NPM1 undergoes liquid-liquid phase separation in the presence of two classes of nucleolar macromolecules: proteins and RNA. Using a multidisciplinary strategy, we identify the structural features that mediate NPM1 phase separation and its nucleoar localization. These results provide a novel perspective on the mechanisms involved in the formation of the liquid-like structure of the nucleolus and in the nucleolar localization of biological macromolecules.

## Results

### NPM1 interacts with proteins displaying multivalent R-rich linear motifs

We first investigated interactions of R-motif-containing proteins with NPM1 by analyzing the results of a whole cell NPM1 pull-down experiment (see Materials and Methods). Of 132 NPM1-binding proteins, 97% exhibited at least one R-motif ($RX_{n_1}R$, where X is any amino acid and $n_1 \leq 2$; *Supplementary file 1*) and 78.8% were annotated with GO terms indicating association with membrane-less organelles (*Supplementary file 2*). Amongst all 132 NPM1-binding proteins, 73% exhibited multiple R-motifs (*Supplementary file 1*); in contrast, only 44% of all human proteins exhibited multiple R-motifs (p<0.0001; *Supplementary file 3*). Thus, multivalent R-motifs are enriched in proteins that bind to NPM1.

### Multivalent R-rich linear motifs mediate phase separation with NPM1

R-motifs bind to a region of NPM1 which includes the OD and a short disordered region (residues 1–130; termed N130; *Figure 2a*). The interactions engage two highly conserved acidic tracts, termed A1 (residues 34–39) and A2 (residues 120–130), at the interface between monomer subunits and

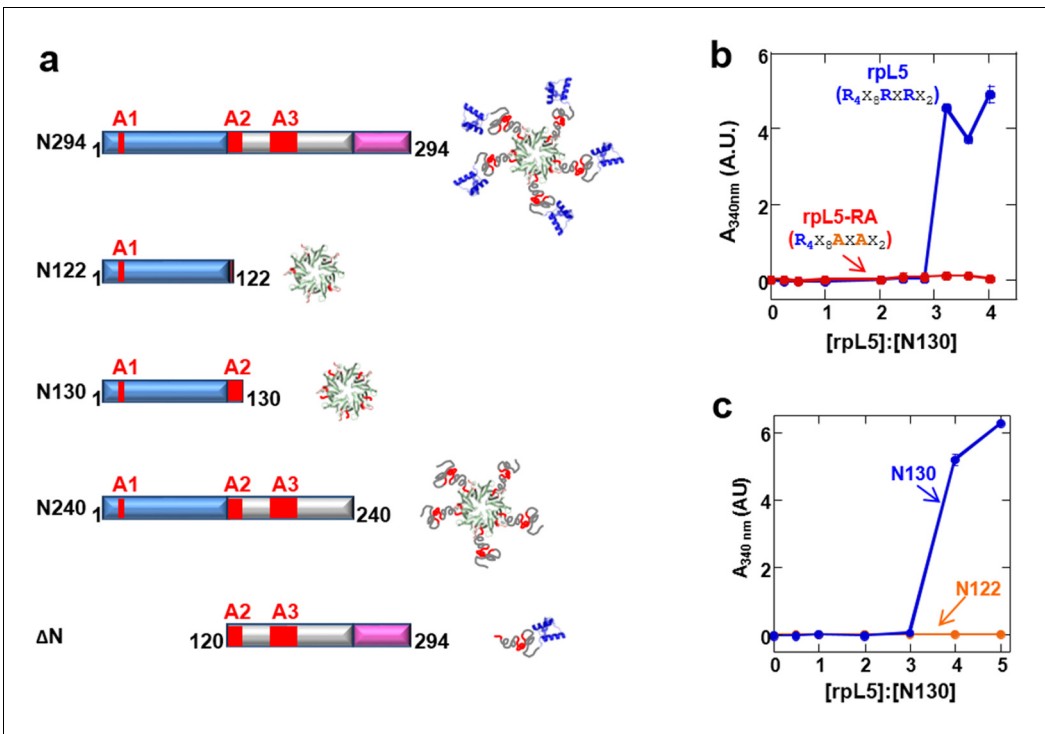

**Figure 2.** Multivalency within both the rpL5 peptide (of R-motifs) and N130 (or A tracts) is required for phase separation. (a) Schematic representation of the NPM1 constructs used in this study; (b) Titrations of rpL5 (blue) or rpL5-RA peptide lacking the second R-motif (red), into 200 µM N130, monitored by light scattering at 340 nm; (c) Titrations of rpL5 peptide into N130 (blue) and N122 (orange).

The following figure supplement is available for figure 2:

**Figure supplement 1.** A monovalent peptide with similar affinity to rpL5 for binding N122 does not phase separate with N130.

within the disordered region (*Mitrea et al., 2014*), respectively (*Figure 2a*). These A tracts and the pentameric nature create multivalency within N130. Multivalent interactions involving low complexity sequences cause assembly and phase separation of biopolymers within membrane-less organelles (*Fromm et al., 2014*; *Li et al., 2012*); therefore, we tested the ability of multivalent N130 and peptides containing multiple R-motifs derived from NPM1-binding proteins to undergo phase separation. Titration of four R-motif-containing peptides (*Table 1*) caused phase separation into liquid-like droplets (*Figure 1b* and *Videos 1–4*) at critical concentrations that varied with R-motif composition and affinity for N130 (*Table 2* and *Figure 1—figure supplement 1*). At 200 µM N130, upon titration of

**Table 1.** Amino acid sequences of the synthetic multivalent R-motif containing peptides.

| Peptide Name[#] | Peptide amino acid sequence | | |
|---|---|---|---|
| | | rpL5 | $^{21}$RRRREGKTDY$_{10}$YARKRLV$^{37}$ |
| GNL2 | $^{682}$RRRAVRQQRP$_{10}$KKVGVRYYET$_{20}$HNVKNRNR$^{709}$ | | |
| SURF6 | $^{299}$RRAQRQRRWE$_{10}$KRTAGVVEKM$_{20}$QQRQDRRR$^{326}$ | | |
| rpL23a | $^{47}$RRPKTLRLRR$_{10}$QPKYPRKSAP$_{20}$RR$^{68}$ | | |
| rpL5-RA | $^{21}$RRRREGKTDY$_{10}$YAAKALV$^{37}$ | | |
| rpL5-2xLinker | RRRREGKTDY$_{10}$YAEGKTDYYA$_{20}$RKRLV | | |

[#]The peptides are referred to by the same name as the protein they originate from and the residue numbers of their N- and C-termini are indicated.

the divalent rpL5 peptide, phase separation was observed when the rpL5:N130 ratio reached ~3:1 (*Figure 2b*). At the same N130 concentration, phase separation was not observed upon titration of a monovalent R-motif peptide (rpL5-RA; *Figure 2b*), even though it bound, albeit with lower affinity (*Table 2* and *Figure 1—figure supplement 1*), confirming that R-motif multivalency is required for phase separation with N130. The inability to phase separate was not due to reduced binding affinity (rather than loss of multivalency); a poly-R peptide, containing a single but longer R-motif, with affinity similar to that of rpL5, also failed to phase separate (*Figure 2— figure supplement 1*). Additionally, phase separation was not observed when an NPM1 construct containing only the OD (N122, residues 1–122; *Figure 2a*) was titrated with the rpL5 peptide (*Figure 2c* and *Table 2*). We thus conclude

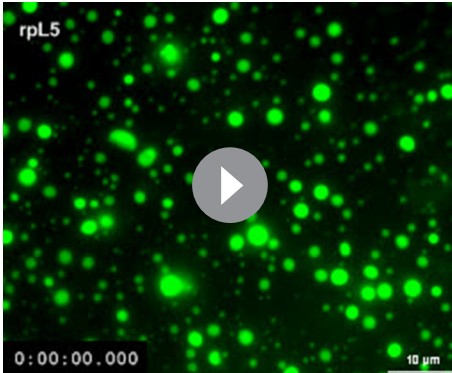

**Video 1.** In vitro droplets formed between 100 µM N130 and 500 µM rpL5. 1 µM N130 was labeled with AlexaFluor488.

that the minimal multivalency requirements for phase separation are the acidic A1 and A2 tracts within NPM1 and at least two complementarily charged R-motifs within a polypeptide binding partner.

## Mutivalent R-peptides mediate inter-pentamer cross-linking in liquid-like droplets

Next, to understand the mechanism of phase separation, we characterized the structural features of complexes of rpL5 with N130 before and after droplet formation. Fluorescence anisotropy (FA) of a N130^S125C mutant, labeled with Alexa Fluor594 within A2, increased in a biphasic manner upon titration of rpL5 (*Figure 3a*). Results from small-angle neutron scattering (SANS) and analytical ultracentrifugation (AUC) experiments showed that monodisperse, soluble complexes formed at the first FA transition, which occurred at ~1:1 rpL5:N130 stoichiometry (*Figure 3a*, inset; *Figure 3a–c*; *Figure 3— figure supplement 1*; *Table 3*). Nuclear magnetic resonance (NMR) spectroscopy showed that rpL5 bound to both the A1 and A2 tracts within $^{15}$N-N130 (*Mitrea et al., 2014*) with a global $K_D$ of 57 ± 14 µM (see Analysis methods) in agreement with isothermal titration calorimetry measurements (*Table 2*). Site-specific $K_D$ values calculated from chemical shift perturbations of individual N130

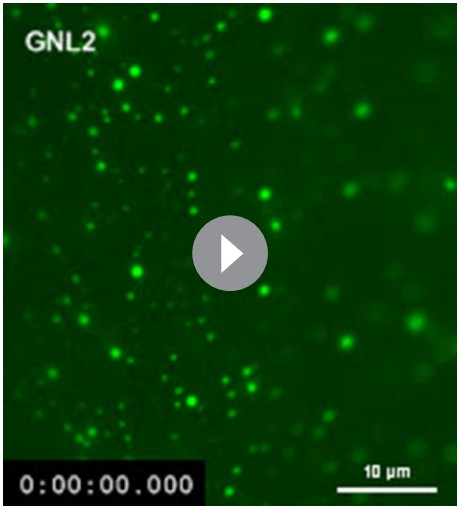

**Video 2.** In vitro droplets formed between 25 µM N130 and 70 µM GNL2. 1 µM N130 was labeled with AlexaFluor488.

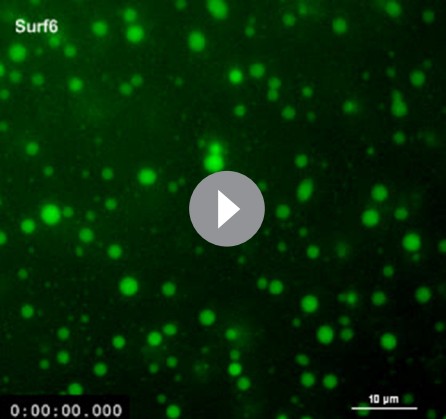

**Video 3.** In vitro droplets formed between 100 µM N130 and 200 µM SURF6. 1 µM N130 was labeled with AlexaFluor488.

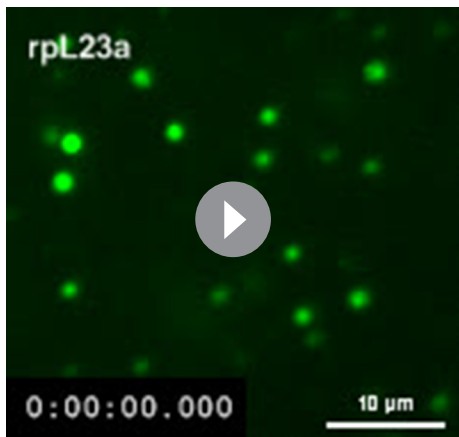

**Video 4.** In vitro droplets formed between 50 µM N130 and 80 µM rpL23. 1 µM N130 was labeled with AlexaFluor488.

peaks upon titration with rpL5 (*Figure 4a*) are presented in *Table 4*. We investigated NMR transverse dipole-dipole/CSA cross relaxation and longitudinal relaxation for backbone amide moieties in $^2$H/$^{15}$N-N130 titrated with rpL5 to determine their dynamic parameters before phase separation (*Figure 4b,c*). Residues within the A2 tract of the apo state, analyzed separately from the folded pentameric core (see Analysis methods), experienced fast, local motions [average local correlation time ($\tau_{c,local}$), 2.10 ± 0.05 ns; average local order parameter ($S_f^2$), 0.44 ± 0.01]; these motions were slowed when N130 was ~93% saturated with rpL5 ($\tau_{c,local}$, 4.56 ± 0.06 ns; *Table 5*) and were reduced in amplitude ($S_f^2$, 0.76 ± 0.01; *Table 5*). Furthermore, a comparison of the overall tumbling time for core residues of N130 in the free and 93% rpL5 saturated state indicated 2:1 binding stoichiometry (see Analysis methods).

The second transition in the FA curve (starting at ~2.5:1 rpL5:N130) corresponded to phase separation (*Figure 3a*). To gain insight into conformational changes associated with phase separation, we performed single-molecule Förster resonance energy transfer ( smFRET) experiments with a mutant of N130, N130$^{Q15C/S125C}$, that could be dually labeled with Alexa Fluor594 & 680 at sites within the pentamer core (Q15C) and the A2 acidic tract (S125C), respectively. The FRET efficiency ($E_{FRET}$) for this dye pair within the droplet phase ($E_{FRET}$ ~0.15) was dramatically reduced in comparison with that observed in the absence of rpL5 ($E_{FRET}$ ~0.85; *Figure 5*), consistent with the extension of the A2 tract from the N130 pentamer core due to interactions with rpL5 molecules upon phase separation (*Figure 6a*). In contrast to the monodisperse character of the SANS curves for samples of rpL5 and N130 at 0:1 and 1:1 (rpL5:N130) stoichiometry (*Figure 3b,c*), the curve for the 1:3 sample indicated periodic structural organization within the liquid-like droplets (*Figure 3b*, red curve). We interpreted these features in terms of inter-molecular correlation distances (55 Å, 77 Å, and 119 Å; see Analysis methods) due to organization of N130 and rpL5 induced by phase separation. Notably, nascent, higher order structural organization was evident in the SANS curve for a solution of rpL5 and N130 with 2:1 stoichiometry (*Figure 3b*, green curve). Assembly of rpL5 and N130 into higher order, soluble intermediates at and above the 2:1 stoichiometric ratio was also demonstrated using sedimentation velocity analytical ultracentrifugation (SV-AUC; *Figure 3—figure supplement 1*, *Table 3*). The

**Table 2.** Binding affinities for interactions between N122 (NPM1 residues 1–122, displaying only A1) and N130 (displaying both A1 and A2) and the multivalent R-motif-containing peptides, determined using isothermal titration calorimetry (ITC), at concentrations below the critical phase separation threshold.

| Peptide | N122 | | N130 | |
| --- | --- | --- | --- | --- |
| | N (sites) | $K_D$ (µM) | N (sites) | $K_D$ (µM) |
| rpL5 | 0.63 ± 0.09 | 19.0 ± 2.7 | 2.18 ± 0.19 | 20.5 ± 5.0 |
| GNL2 | 0.66 ± 0.04 | 10.2 ± 1.3 | n.d.[*] | n.d.[*] |
| SURF6 | 0.55 ± 0.06 | 9.2 ± 0.5 | 1.4 ± 0.02 | 19.0 ± 0.8 |
| rpL23a | n.d.[#] | n.d.[#] | n.d.[*] | n.d.[*] |
| rpL5-RA | 1.0 ± 0.06 | 50.9 ± 3.18 | 0.84 ± 0.07 | 140.1 ± 9.29 |

[*] Not determined due to multiple, unresolved binding events

[#] Heat change was too weak for accurate data analysis

Average values from a minimum of three independent experiments are reported ± SD

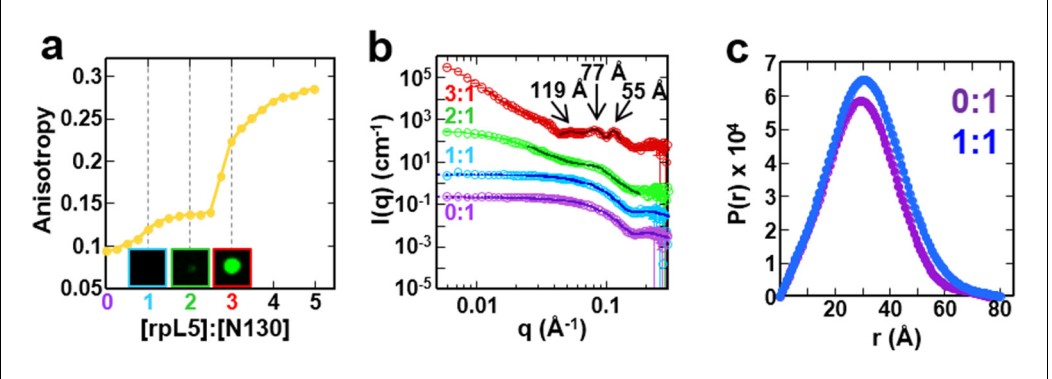

**Figure 3.** The liquid-like phase formed by rpL5 and N130 is characterized by molecular ordering and is accompanied by soluble, oligomeric intermediates. (a) Fluorescence anisotropy of Alexa Fluor594-labeled N130 (at S125C within acidic tract A2; 1 µM) upon titration of the rpL5 peptide; the total [N130] was 200 µM. Insets: light microscopy images of the 1:1 (cyan box), 2:1 (green box) and 3:1 (red box) rpL5:N130 solutions. The same stoichiometry color coding is used in all panels. (b) SANS curves, $I(q)$ versus $q$, for rpL5:N130 (200 µM) solutions at 0:1, 1:1, 2:1, and 3:1 stoichiometry. The curve for the 0:1 solution is on the absolute $I(q)$ scale (cm-1) with the others shifted in 1, 2 and 4 decade increments for clarity. Fits (solid lines) of the curves to obtain $R_g$ values (0:1 and 1:1 solutions) and correlation distances (2:1 and 3:1 solutions) are shown (See Analysis methods for details on curve fitting). (c) Pair-distribution, $P(r)$, curves for 0:1 and 1:1 rpL5:N130 were calculated from the corresponding SANS curves (fits shown in *Figure 3b*). For apo N130, the $D_{max} \sim 78.65$ Å with a resulting $R_g = 23.04 \pm 0.09$ Å and $I(0) = 0.2344 \pm 0.0006$ cm-1. From $I(0)$ and using Eq. S1, the estimated molecular mass, $M = 76$ kDa, was determined; this mass is consistent with the expected five subunits within the pentamer (subunit $M = 14.6$ kDa). For 1: 1 rpL5:N130, the $D_{max} \sim 80.32$ Å with a resulting $R_g = 24.3 \pm 0.1$ Å and $I(0) = 0.2744 \pm 0.0009$ cm-1. Here, $I(0)$ yields $M = 89$ kDa, indicative of ~5 rpL5 ($M = 2.2$ kDa) molecules bound to N130.

The following figure supplement is available for figure 3:

**Figure supplement 1.** Sedimentation velocity analytical ultracentrifugation (SV-AUC) profiles for N130 titrated with rpL5.

appearance of high molecular weight species within the detection range (<1 MDa) was accompanied by a progressive decrease in the total mass detected, likely due to sedimentation of rpL5:N130 droplets in the sample cell during the dead-time of the experiment. Above the phase separation threshold (>3:1 rpL5:N130), resonances for residues within the N130 core broadened beyond detection in 2D $^1$H-$^{15}$N TROSY spectra but not those for residues within the A2 tract (*Figure 4a*). Chemical shift values indicated that these residues remained disordered.

Integrating our structural results, we propose a model of rpL5/N130-dependent phase separation, as follows (*Figure 6b*). As rpL5 is titrated into N130 up to 2:1 stoichiometry (rpL5:N130), the R1 motifs, comprised of four Arg residues each, bind to acidic residues within the A1 binding groove and disordered A2 tract of N130, with the R2 motif available for interactions. Upon further titration of rpL5, at the critical phase separation concentration, when the higher affinity sites reach a critical saturation threshold, R2 motifs within rpL5 molecules already bound to N130 pentamers transiently engage A tracts of other pentamers, possibly within the longer and disordered A2 tract, establishing inter-N130 pentamer cross-links. Together, our data support the hypothesis that the molecular basis of phase separation is the formation of non-covalent, inter-N130 pentamer interactions via the two R-motifs within the same rpL5 peptide molecule. We propose that these rpL5-mediated interactions establish the inter-pentamer spacing within the droplet phase that was detected by SANS (*Figure 3b*). In order to validate this model, we synthesized a variant rpL5 peptide in which the eight residue-long linker connecting the two R-motifs (see *Table 1*) was duplicated. The phase separated sample formed between the peptide with the longer linker (rpL5-2xLinker) exhibited altered intermolecular spacing, as indicated by altered correlation distances derived from the SANS curve (*Figure 7*). These shifted to positions corresponding to larger correlation distances, in agreement with the hypothesis that the R-motif peptides establish inter-N130 pentamer spacing in the liquid-like

**Table 3.** Results of sedimentation velocity analytical centrifugation analysis (SV-AUC) of N130 in the absence or presence of increasing concentrations of the rpL5 peptide.

| rpL5:N130 | mg/ml[a] | s$_{20}$ (Svedberg)[b] | s$_{20,w}$ (Svedberg)[c] | MW (kDa)[d] | f/f$_0$ [e] |
|---|---|---|---|---|---|
| 0:1 | 2.76 | 4.63 (98.7%) | 4.82 | 88.9 | 1.53 |
| | | 7.90 (1.20%) | 8.22 | 198.0 | 1.53 |
| 1:1 | 3.09 | 5.23 (96.3%) | 5.44 | 99.3 | 1.46 |
| | | 7.63 (3.7%) | 7.94 | 175.0 | 1.46 |
| 2:1 (1-300 min) | 3.63 | 1.07 (2.6%) | 1.11 | 6.8 | 1.20 |
| | | 5.80 (48%) | 6.03 | 86.3 | 1.20 |
| | | 8.11 (23%) | 8.44 | 143.0 | 1.20 |
| | | 12.33 (25%) | 12.82 | 268.0 | 1.20 |
| | | 14-19 (7%) | 14-19 | 300-500 | 1.20 |
| 2:1 (100-300 min)[f] | 3.31 | 5.75 (56%) | 5.98 | 81.1 | 1.16 |
| | | 5.80 (25%) | 6.03 | 128.0 | 1.16 |
| | | 8.11 (17%) | 8.44 | 182.0 | 1.16 |
| 3:1 (1-300 min) | 2.01 | 0.95 (17%) | 0.98 | 7.4 | 1.43 |
| | | 5.80 (31%) | 6.00 | 112.0 | 1.43 |
| | | 10.32 (32%) | 10.73 | 266.0 | 1.43 |
| | | 14-27 (21%) | 14-27 | 500-850 | 1.43 |
| 3:1 (100-300 min)[f] | 1.29 | 1.18 (4%) | 1.22 | 1.1 | 1.49 |
| | | 5.73 (41%) | 5.96 | 117.0 | 1.49 |
| | | 7.41 (20%) | 7.71 | 173.0 | 1.49 |
| | | 9.17 (15%) | 9.54 | 238.0 | 1.49 |
| | | 11.33 (15%) | 11.79 | 266.0 | 1.49 |

[a] Total concentration in mg/ml.

[b] Sedimentation coefficient taken from the ordinate maximum of each peak in the best-fit c(s) distribution at 20 °C with percentage protein amount in parenthesis. Sedimentation coefficient (s-value) is a measure of the size and shape of a protein in a solution with a specific density and viscosity at a specific temperature.

[c] Standard sedimentation coefficient (s$_{20,w}$-value) in water at 20 °C.

[d] Molar mass values (MW) taken from the c(s) distribution that was transformed to the c(M) distribution. The theoretical molar mass is in parenthesis.

[e] Best-fit weight-average frictional ratio values (f/f$_0$)$_w$ taken from the c(s) distribution.

[f] Fit of the data in the 100-300 min sedimentation time frame, for the analysis of the lower MW rpL5:N130 complex intermediates. The MW cutoff of ~ 300 kDa corresponds to a tetramer of N130 pentamers.

phase. We envision that such cross-links are dynamically formed and broken but that, once phase separation occurs, they dominate the structural organization detected by SANS.

## NPM1 incorporates both R-rich peptides and rRNA into multicomponent liquid-like droplets

Our data showed that the minimal construct, N130, supported droplet formation. However, both the OD (*Enomoto et al., 2006*; *Jian et al., 2009*) and NBD (*Hisaoka et al., 2014*; *Negi and Olson, 2006*) are required for nucleolar localization of transfected NPM1, suggesting that interactions with ribosomal RNA (rRNA) are also involved in integration of NPM1 within the GC matrix. Full length NPM1 also exhibits a central intrinsically disordered region (IDR) that, in addition to the A2 tract, contains a longer acidic tract termed A3 (*Figure 1a*). We tested and confirmed the ability of full length NPM1 (N294) to phase separate with either wheat germ rRNA or rpL5 using light scattering (*Figure 8a & b*, respectively) and fluorescence microscopy (*Figure 9a*, Rows 1 & 2, respectively). Binary mixtures of N294 with either rpL5 or rRNA formed droplets that increased in size over time (*Figure 9b*); those with rpL5 fused more rapidly and were larger in size than those with rRNA.

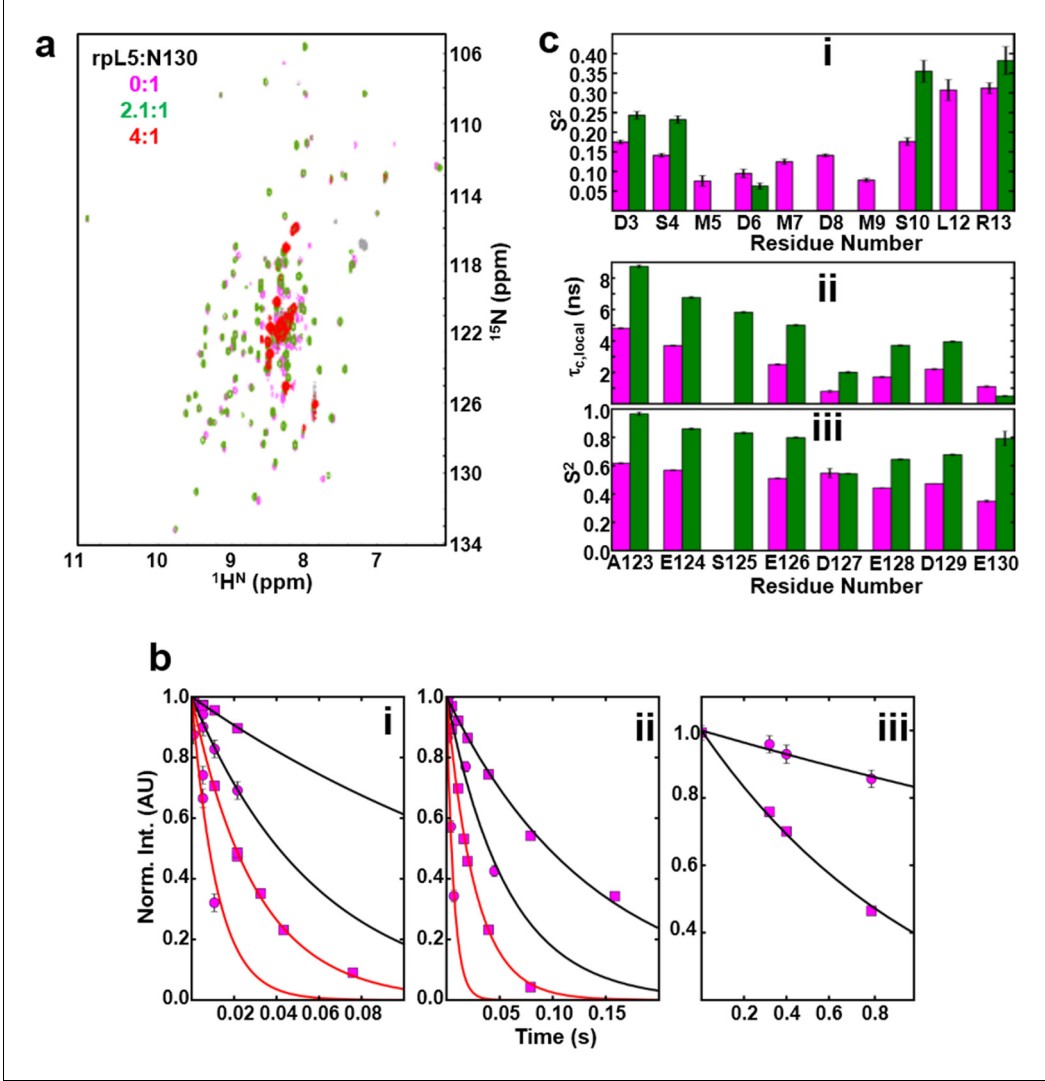

**Figure 4.** NMR chemical shift perturbations and amide backbone relaxation analysis of apo N130 and soluble complexes at 2.5:1 rpL5:N130 stoichiometry. (a) $^1$H-$^{15}$N TROSY-HSQC spectra of apo N130 (magenta), in a soluble complex with rpL5 (2.1:1 rpL5:N130; green) and in the phase separated state (4:1 rpL5:N130; red). (b) Examples of $R_{2,\beta}$ and $R_{2,\alpha}$ peak intensity decay curves measured at 800 MHz (i) and 1000 MHz (ii), and $R_1$ peak intensity decay curves measured at 800 MHz (iii) for free N130. In (i) and (ii), points and curves in black and red correspond to the $R_{2,\beta}$ and $R_{2,\alpha}$ experiments, respectively. In (i)–(iii), circles and squares indicate data points for residues Tyr29 (within the folded pentamer core) and Glu121 (within the disordered A2 tract), respectively. Solid lines correspond to fits using a simple exponential decay model from which relaxation rates and intensities at zero time were extracted. (c) (i) Comparison of S2 values for the N-terminus for which data could be collected for apo N130 (magenta bars) and 2.5:1 rpL5:N130 (green bars). Comparison of $\tau_{c,local}$ (ii) and S2 (iii) values for C-terminal residues in apo N130 (magenta bars) and 2.5:1 rpL5:N130 (green bars). S125 is overlapped in the apo N130 spectrum; therefore, under this condition, these analyses could not be performed.

Furthermore, droplets of NPM1 with rpL5 formed at ~5-fold lower concentrations of both components (*Figure 8b*) than were required for phase separation with N130 and rpL5 (*Figure 1b*) due to the higher valency of acidic tracts within the full-length protein. Interestingly, rpL5 (and other R-motif peptides) and rRNA phase separated in the absence of N294 (*Figure 8c,d*), forming very small puncta (*Figure 9a*, Row 3). We cannot explain the physical basis for the different dynamics of liquid-like droplets formed by N294 with either rRNA or rpL5 (*Figure 9a*, Rows 1 and 2, respectively) or the morphology of the punctate structures formed by rRNA and rpL5 (*Figure 9a*, Rows 3) due to the lack of data on the interaction between NBD of NPM1 or rpL5, and rRNA (*i.e.*, binding affinities,

**Table 4.** NMR-derived dissociation constant ($K_D$) values determined by monitoring chemical shift perturbations of individual N130 nitrogen backbone resonances while titrating the rpL5 peptide.

| Residue | Structural context | $K_D$ (µM) |
|---|---|---|
| Met5 | N-terminus | 385 ± 108 |
| Asp6 | N-terminus | 116 ± 25 |
| Met7 | N-terminus | 142 ± 30 |
| Ser10 | N-terminus | 56 ± 13 |
| Leu12 | N-terminus | 10 ± 5 |
| Arg13 | N-terminus | 58 ± 13 |
| Gln15 | Core | 36 ± 10 |
| Tyr17 | Core | 19 ± 9 |
| Leu18 | Core | 205 ± 51 |
| Val33 | Core | 1041 ± 649 |
| Asp36 | A1 tract | 329 ± 93 |
| Glu37 | A1 tract | 1549 ± 1168 |
| Glu39 | A1 tract | 115 ± 37 |
| His40 | Core | 123 ± 23 |
| Leu42 | Core | 24 ± 9 |
| Ser43 | Core | 166 ± 38 |
| Ala64 | Core | 1227 ± 562 |
| Asn66 | Core | 1530 ± 958 |
| Tyr67 | Core | 245 ± 58 |
| Glu68 | Core | 147 ± 29 |
| Val74 | Core | 53 ± 14 |
| Phe92 | Core | 14 ± 6 |
| Glu93 | Core | 329 ± 78 |
| Ile94 | Core | 73 ± 17 |
| Thr95 | Core | 34 ± 9 |
| Leu116 | Core | 101 ± 24 |
| Val117 | Core | 61 ± 17 |
| Ala118 | Core | 35 ± 9 |
| Glu120 | A2 tract | 68 ± 16 |
| Glu121 | A2 tract | 101 ± 18 |
| Asp122 | A2 tract | 279 ± 76 |
| Ala123 | A2 tract | 182 ± 50 |
| Glu124 | A2 tract | 614 ± 410 |
| Glu126 | A2 tract | 576 ± 471 |
| Asp127 | A2 tract | 2162 ± 1612 |
| Glu130 | A2 tract | 4258 ± 1986 |

number and location of binding sites, etc.). The size difference between the two types of droplets with N294 (*Figure 9a*, Rows 1 and 2) may, however, arise from differences in binding affinity between rRNA and NBD *versus* rpL5 and A tracts within OD/IDR.

Intrigued by the physical differences between the structures formed by pairwise combinations of N294, rpL5 and rRNA, we next sought to understand the behavior of these three species in ternary mixtures by examining interactions between pre-formed droplets comprised of N294 and rRNA to which freely diffusing rpL5 was added, at a concentration below that which caused phase separation

**Table 5.** NMR-derived dynamic parameters for monodisperse apo N130 and a 2.5:1 rpL5:N130 soluble complex.

| | Apo N130 (N-terminus/A2 tract) | 2.5:1 rpL5:N130 (N-terminus/A2 tract) |
|---|---|---|
| $\tau_{c,local}$(ns) | $-^a$ / 2.10 ± 0.05 | $-^a$ / 4.56 ± 0.06 |
| $S_f^2$ | 0.16 ± 0.01$^a$ / 0.44 ± 0.01 | 0.26 ± 0.02$^a$ / 0.76 ± 0.01 |

$^a$Dynamic parameters for the N-terminus were determined using a global $\tau_c$ value; thus, the only residue-specific motional parameter for residues within the N-terminus was $S_f^2$ (see Materials and Methods).

with N294 alone (*Figure 9a*, Row 4). The peptide accumulated into the rRNA/N294 droplets, highlighting NPM1's capacity to bind the two fundamental classes of macromolecules present in the nucleolus; these droplets also slowly grew over time (*Figure 9b*). Importantly, this multi-modal binding mediates the co-assembly of rRNA and rpL5 within a dense, multi-component liquid-like phase. We hypothesize that a similar molecular mechanism is responsible for the phase separation and co-localization of nucleolar components within the GC.

We next examined the roles of the different domains of NPM1 in phase separation to form multi-component liquid-like droplets using two truncation mutants, one lacking the NBD (N240, residues 1–240; *Figure 2a*) and another lacking the OD (ΔN, residues 120–294; *Figure 2a*). In agreement with a mechanistic model wherein multivalent interactions between pentameric A1, A2, and A3 tracts within NPM1 and multivalent R-motifs within its nucleolar protein partners mediate phase separation, the N240, but not the ΔN construct phase separated with rpL5 (*Figure 10a*). Furthermore, neither of these truncated constructs experienced phase separation in the presence of rRNA, confirming that multivalent display of the NBD is required for the co-localization of rRNA with NPM1 within liquid-like droplets (*Figure 10b*).

Multi-modal binding to two classes of macromolecules, R-motif-containing nucleolar proteins (binding mode 1) and rRNA (binding mode 2), is likely critical for NPM1-dependent formation of multi-component liquid-like droplets. In the absence of NPM1, rRNA and rpL5, representing these two classes, phase separated into small puncta (*Figure 8c,d*, *Figure 9*). Given this result, we next asked whether addition of NPM1 constructs to these puncta would cause reorganization and co-localization of the constituent macromolecules within larger, liquid-like droplets. In order to differentiate between the effects of interactions between the OD/A tracts and rpL5 (mode 1) and OD/NBD and rRNA mode (mode 2) on phase separation, we first formed rRNA/rpL5 puncta at two concentrations of rpL5 and then added the NPM1 constructs and monitored phase separation using confocal-microscopy. At the high rpL5 concentration (rpL5, 200 μM; NPM1 constructs, 30 μM; termed the 'excess' condition), rpL5 and NPM1 could independently phase separate. However, at the low concentration (rpL5, 50 μM; NPM1 constructs, 30 μM; termed the 'limiting' condition), rpL5 and NPM1 could not independently phase separate. The addition of N294 to pre-formed, rpL5/rRNA puncta caused spontaneous formation of large droplets under conditions of both limiting and excess rpL5, whose sizes increased with increasing N294 concentration (*Figure 11*). In contrast, addition of the N240 deletion construct, lacking the NBD, caused phase separation under conditions of excess rpL5, but not with a limiting amount of rpL5 (*Figure 11*). Finally, addition of the ΔN construct, lacking the OD and therefore displaying dramatically reduced multivalency, failed to cause phase separation under both excess and limiting rpL5 conditions (*Figure 11*). Together, these results support a mechanism wherein the high valency associated with the NPM1 OD, together with the multiple acidic tracts and NBD, are required for the dissolution of rpL5/rRNA puncta and co-localization of these molecules within large liquid-like droplets that readily grow in size. We thus propose that, through multi-modal interactions with two major classes of nucleolar macromolecules, NPM1 localizes to the nucleolus and also mediates the co-localization of other protein-binding partners and rRNA within the GC of the nucleolus.

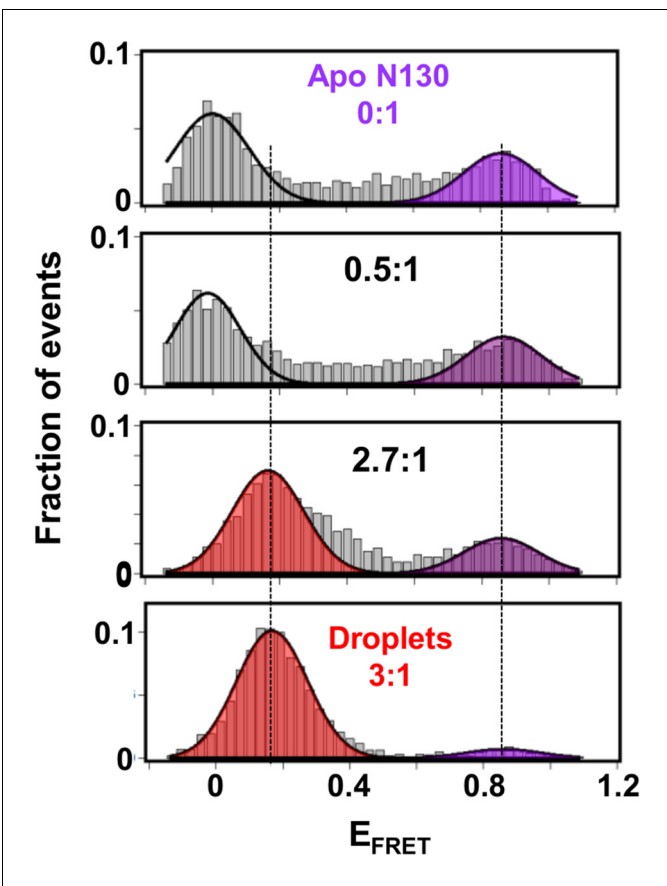

**Figure 5.** The A2 tract extends away from the folded core of N130 on the pathway to phase separation. smFRET histograms for Alexa Fluor594/680-labeled N130$^{Q15C/S125C}$ (~100 pM; total [N130], 200 µM) at 0:1, 0.5:1, 2.7:1 and 3:1 (phase separated state) rpL5:N130 stoichiometry; the FRET events corresponding to conformational states with similar $E_{FRET}$ values are indicated by the shaded regions. The solid lines are Gaussian fits of the data, from which average $E_{FRET}$ values were determined.

## R-motif and nucleic acid binding by pentameric NPM1 are both required for nucleolar localization

We next explored the relevance of our in vitro findings to nucleolar physiology. First, we monitored the accumulation of a series of recombinant eGFP-fused NPM1 constructs within isolated nucleoli (*Figure 12a*) using confocal fluorescence microscopy. Consistent with in vitro results showing that all domains of NPM1 were required for the integration of both an R-motif peptide and rRNA into large, liquid-like droplets, the eGFP-N294 fusion protein accumulated within purified nucleoli, even at sub-micromolar concentrations (*Figure 12b,c*), while constructs lacking either the rRNA binding or oligo-merization domains (eGFP-N240 or eGFP-△N, respectively) did not (*Figure 12c*). These results suggested that the nucleolar localization of NPM1 requires multi-modal interactions with both R-motif-containing nucleolar proteins and rRNA.

To extend our studies to the cellular setting, we next investigated the domain requirements for nucleolar localization by creating *Trp53$^{-/-}$/Npm1$^{-/-}$* mouse embryonic fibroblasts cell lines (DKO MEFs) that stably expressed a related series of mCherry-fused NPM1 constructs. The parental DKO cell line was created by replacing exons 2–7 of *Npm1* with *eGFP*, and therefore constitutively expresses the fluorescent protein (*Grisendi et al., 2005*). The stable DKO MEF cell lines were created through infection with retroviruses carrying the Thy1.1 cell surface marker in frame with an internal ribosomal entry site (IRES) and an mCherry-NPM1 variant gene. The two genes are translated from the same bicistronic mRNA transcript at similar levels (*Gurtu et al., 1996*). Therefore, the Thy1.1 cell surface marker is an unbiased reporter of mCherry-NPM1 variant expression. mCherry is

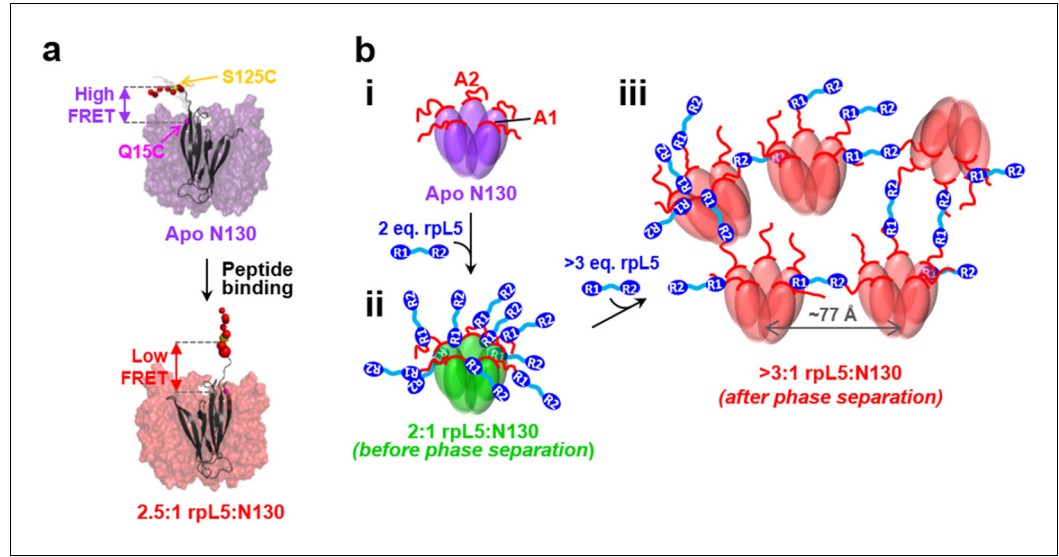

**Figure 6.** Schematic model of the structural rearrangements in N130 associated with liquid-liquid phase separation in the presence of the multivalent rpL5 peptide. (**a**) Structural representation of results from NMR and smFRET that revealed dramatic changes in dynamics and spatial orientation of the A2 track of N130 upon rpL5 peptide binding: apo N130 (top) and N130 in complex with rpL5 (bottom). Residues within the A2 tract of N130 are shown as colored spheres with diameters proportional to the $\tau_{c,local}$ value for the $^{15}$N nucleus of amide group. The sites of fluorescent labeling, Q15C and S125C, are indicated as purple and yellow spheres, respectively (the spheres for Q15C do not encode dynamic information). In the apo state, A2 randomly samples relatively compact conformations, while in liquid-like droplets, these residues extend away from the N130 core. (**b**) Schematic representation of apo N130 (**i**) and the proposed structural model of phase separation. Upon saturation of the two principal binding sites within the A1 and A2 tracts on N130 (**ii**), bound rpL5 peptides extend toward and engage in weak interactions with neighboring N130 pentamers (**iii**), thus creating 3D, expandable cross-links. In (**iii**), only a subset of the possible inter-N130 pentamer crosslinks are shown for clarity. We suggest that one of the correlation distances observed using SANS (77 Å) corresponds to the inter-N130 pentamer spacing.

primarily localized to the cytoplasm and is stable in cells (*Shaner et al., 2004*); therefore, with the constructs under study here, any alteration of its localization and stability can be attributed to its NPM1 variant fusion partner. In agreement with the *in vitro* and *ex cellulo* results, full length mCherry-NPM1 accumulated in nucleoli and co-localized with the nucleolar markers NOPP140 (*Figure 12d*) and fibrillarin (*Figure 12—figure supplement 1*). Despite the fact that all cell lines expressed comparable levels of the Thy1.1 marker (*Figure 12—figure supplement 2*), and all NPM1 constructs encode the native bi-partite nuclear localization signal (residues 152–157 and 191–197 [*Hingorani et al., 2000*]), mCherry fluorescence was undetected in the mCherry-N240 cell lines, while the mCherry-ΔN cell lines exhibited diffuse mCherry fluorescence throughout the nucleus (*Figure 12d* and *Figure 12—figure supplement 1*). Notably, Enomoto, *et al.,* previously showed that NPM1 mutants which lacked the ability to accumulate within nucleoli exhibited dramatically reduced half lives in cells (*Enomoto et al., 2006*), a potential explanation for the undetectable levels of mCherry-N240 in our study. We note that our observations through use of DKO cell lines contrast with those of others (*Enomoto et al., 2006*; *Negi and Olson, 2006*) wherein NPM1 constructs lacking the C-terminal domain accumulated within nucleoli. These previous studies are confounded by the possiblility for the formation of heteromeric oligomers comprised of endogenous wild-type and mutant NPM1 proteins. Collectively, these data suggest that nucleolar localization of NPM1 is achieved only when interactions with both R-motif-containing nucleolar proteins and rRNA are possible. Deletion of either the NBD or OD abrogated the ability of NPM1 to simultaneously interact with an R-motif-containing peptide and rRNA within droplets *in vitro* (*Figure 11*), suggesting that co-localization of NPM1 with the two types of nucleolar components in dense liquid-like droplets and localization within nucleoli arise through similar mechanisms involving multi-modal, multivalent interactions that promote phase separation.

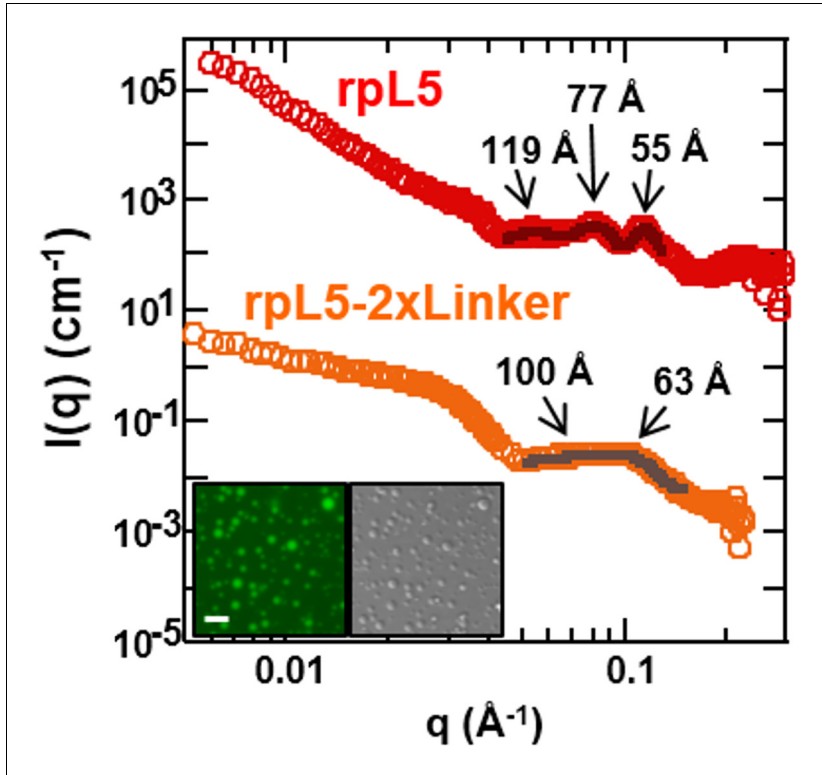

**Figure 7.** The length of the linker between R-motifs in the rpL5 peptide influences the molecular organization within rpL5:N130 liquid-like droplets. (a) SANS curves, *I(q) versus q*, for 3:1 rpL5:N130 (red) and rpL5-2xLinker:N130 (orange) solutions; the N130 concentration for both was 200 μM. The curve for the droplets containing rpL5 2xLinker is displayed on absolute *I(q)* scale (cm$^{-1}$) with the curve corresponding to rpL5 containing droplets shifted in 4 decade increments for clarity. Fits (solid lines) of the curves to obtain correlation distances are shown (See Analysis methods for details on curve fitting). Peak positions for the droplets with the rpL5-2xLinker correspond to correlation distances (*d*) of 63 Å & 100 Å whereas those observed with the wild-type rpL5 peptide correspond to *d* values of 56 Å, 79 Å & 119 Å. *Inset:* wide-field fluorescence microscopy (left) and DIC (right) images of droplets formed from the 3:1 rpL5-2xLinker:N130 solution. Scale bar = 10 μm.

## Discussion

The nucleolus is a membrane-less organelle comprised of a vast array of macromolecules that mediate ribosome biogenesis and participate in stress signaling. However, the molecular mechanism that underlies the localization of macromolecules within the nucleolus is poorly understood. In contrast to the nuclear localization process, for example, which relies on nuclear transport receptors that recognize their cargo via a specific short linear motif (*Christie et al., 2015*) and transport them across the nuclear membrane through the nuclear pore complex, there are no known molecular transporters that mediate localization of proteins to the nucleolous. It has been hypothesized that enrichment of proteins within the nucleolus occurs though selective retention within the nucleolar matrix. Short basic sequences, termed nucleolar localization signals (NoLS), often rich in R and lysine (K) residues, have been identified experimentally to be necessary and, in some cases, sufficient for localization of certain proteins within the nucleolus (*Scott et al., 2010*). However, the molecular basis for how these canonical signals cause nucleolar localization is presently unknown. Here, we identified multivalent, low complexity R-motifs within proteins that are associated with binding to the abundant nucleolar protein, NPM1. Furthermore, we showed that these multivalent R-motifs underwent phase separation with NPM1 to form dense, liquid-like droplets, and became co-localized within these types of droplets that also contained rRNA. Strikingly, the multivalent R-motifs that we identified to mediate interactions with NPM1 have features in common with the canonical NoLSs: they are enriched in basic amino acids and are often located within disordered and solvent accessible regions of proteins

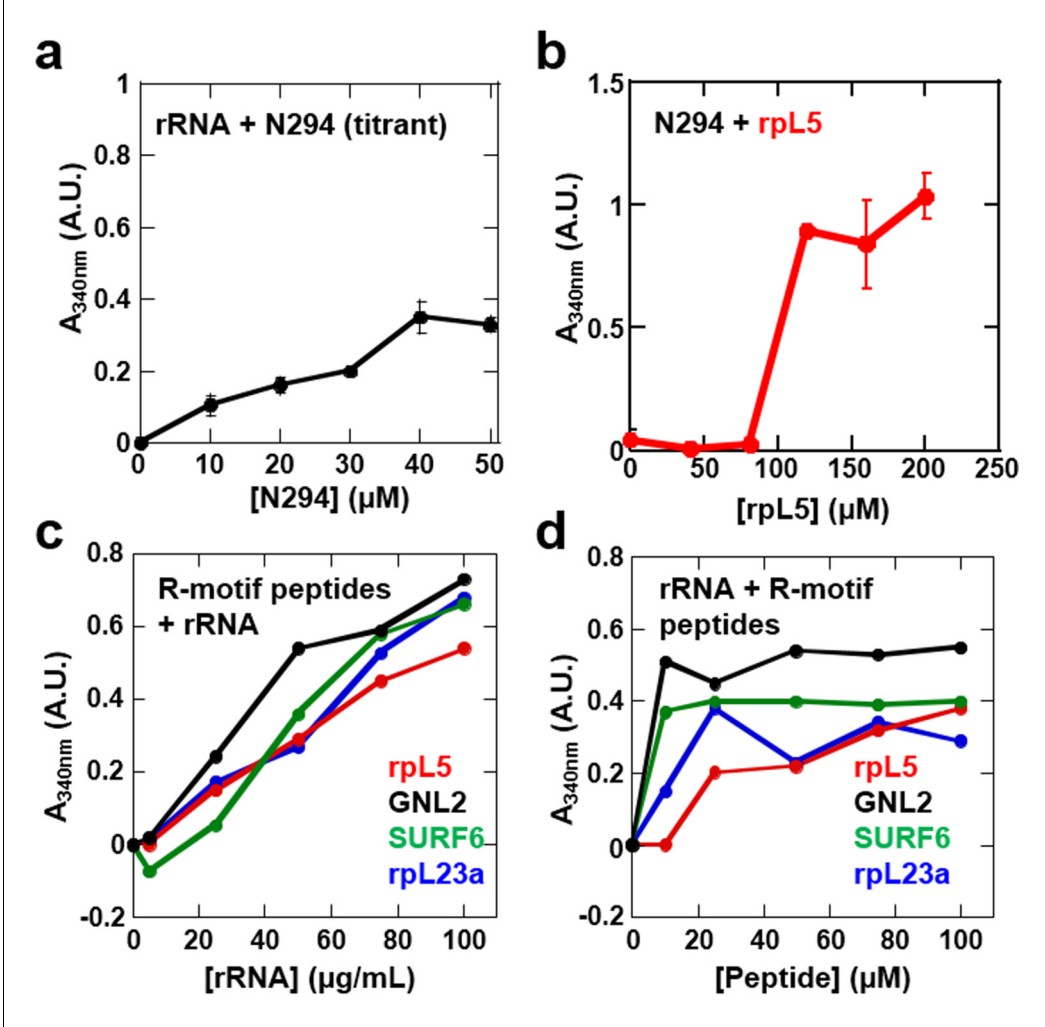

**Figure 8.** Phase separation by pairwise mixtures of NPM1 (N294) and molecules representing two fundamental components of the nucleolus, R-motif containing proteins (represented by R-motif peptides) and rRNA, as determined by light scattering assays. Titration of (a) N294 into 50 μg/mL rRNA and (b) rpL5 into 40 μM N294; (c) Titrations of rRNA into 50 μM R-motif containing peptides (rpL5, GLN2, SURF6, and rpL23a). (d) Titrations of R-motif containing peptides into 50 μg/mL rRNA (as in c).

(*Scott et al., 2010*) (*Figure 1—figure supplement 2*). This observation, however, raises several questions. Do the links between canonical NoLSs and nucleolar localization, and multivalent R-motifs and NPM1 binding, have a common mechanistic basis? And, does nucleolar localization rely on the ability of proteins to experience phase separation within a multi-component, liquid-like phase? To address these questions, we compared our list of NPM1 interacting proteins that contained multivalent R-motifs with the database of all predicted NoLSs within the human proteome ([*Scott et al., 2010*]; http://www.compbio.dundee.ac.uk/www-nod/downloads/AllPredictedHumanNoLSs.txt). Interestingly, 83 (63%) of the proteins from our curated list of NPM1 interactors corresponded to those with predicted NoLSs. Of these 83, 69 (83%) contained 2 or more R-motifs, and there was extensive overlap between the sequence locations of the multivalent R-motifs that we identified and those of the predicted NoLSs (*Supplementary file 1*). Based on these observations, we propose that canonical NoLSs, which closely match our definition of multivalent R-motifs, enable proteins that are freely diffusing through the nuclear compartment to phase separate within the nucleolar matrix, thereby mediating their nucleolar localization.

Furthermore, while the NoLSs described by Scott, *et al.*, account for the nucleolar localization of hundreds of proteins, more than 4,500 proteins are known to reside within the nucleolus

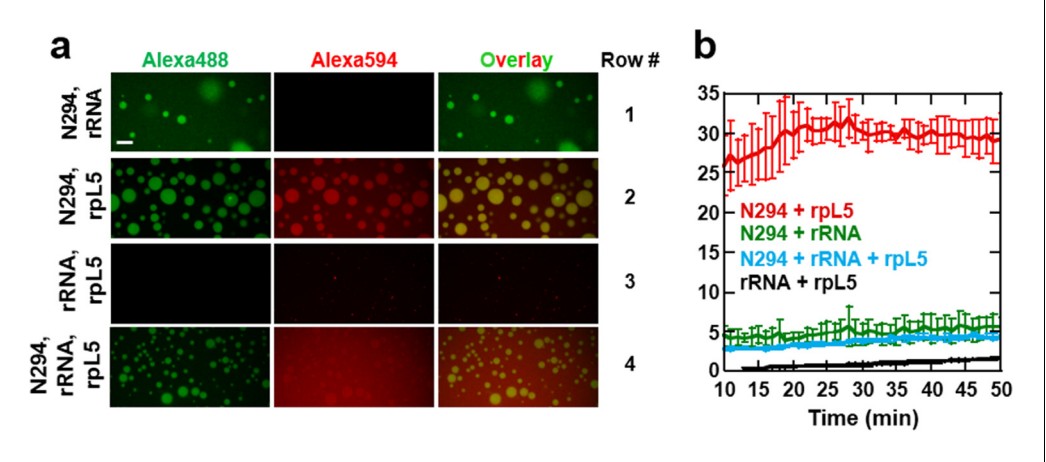

**Figure 9.** Multi-modal binding of NPM1 mediates formation of liquid-like droplets with both rRNA and rpL5 in vitro. (**a**) Confocal microscopy images of droplets after 15 min incubation formed between 40 µM N294 and 50 µg/mL rRNA (Row 1), 40 µM N294 and 200 µM rpL5 (Row 2), 200 µM rpL5 and 50 µg/mL rRNA (Row 3) and 40 µM N294 and 50 µg/mL rRNA droplets treated with 40 µM rpL5, a concentration below the critical phase separation threshold for N294/rpL5 binary system (Row 4). rpL5 and N294 were labeled with Alexa Fluor594 and Alexa Fluor488, respectively. Scale bar = 10 µm; (**b**) Quantitation of droplet size over time for the samples presented in panel **a**; values plotted are the mean ± SD, n = 4 fields of view.

(*Ahmad et al., 2009*), highlighting the need to identify additional localization mechanisms. Interestingly, along these lines, NPM1, although well established to be highly enriched in the nucleolus, does not exhibit a canonical NoLS. Rather, here we describe a multi-modal interaction model that enables the multifunctional protein NPM1 to interact with R-motif-containing proteins and rRNA, representative of two major classes of macromolecules resident of the nucleolus. In support of this, we found that disruption of its multi-modal interactions, through abrogation of NBD oligomerization by the deletion of the OD or removal of the NBD, hindered NPM1's ability to phase separate *in vitro* and to accumulate in nucleoli within mammalian cells (*Figures 10–12*). Consistent with our model, a

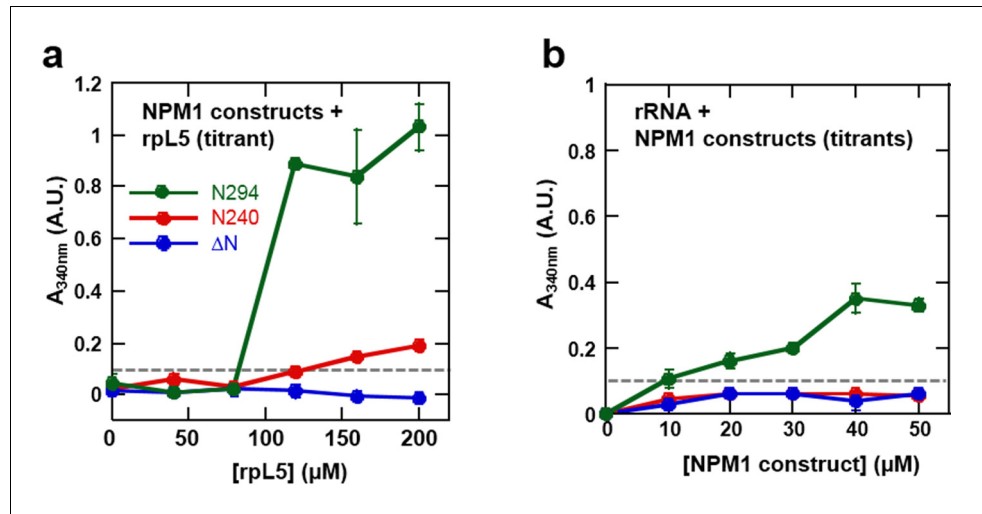

**Figure 10.** The OD and A tracts of NPM1 are required for phase separation with rpL5, while phase separation in the presence of rRNA requires both folded domains (OD & NBD). Light scattering assays of (**a**) titrations of rpL5 peptide into 40 µM NPM1 and (**b**) titrations of NPM1 constructs into 50 µg/mL rRNA. Values plotted are the mean ± SD, n = 3 experiments. The dashed line at 0.1 AU indicates the threshold, above which visible turbidity and microscopic droplets can be detected.

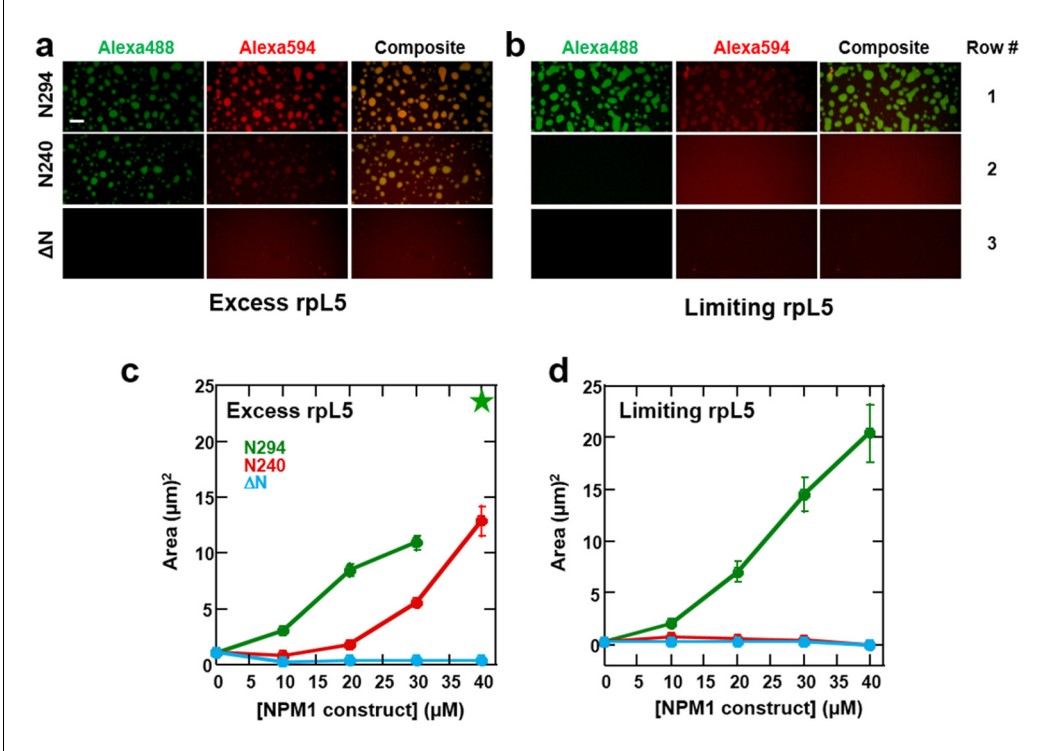

**Figure 11.** Both the OD and NBD of NPM1 are required for co-localization of rRNA and rpL5 within liquid-like droplets in vitro. Confocal microscopic images 15 min after the addition of the specified NPM1 construct (30 μM) to preformed rpL5/rRNA puncta formed from 50 μg/mL rRNA and 200 μM rpL5 (a) and from 50 μg/mL rRNA and 50 μM rpL5 (b). Scale bar = 10 μm. Quantification of the growth of droplets formed through disassembly of rpL5/rRNA puncta upon addition of individual NPM1 constructs. (c) NPM1 constructs were titrated into puncta formed between 200 μM rpL5 and 50 μg/mL rRNA, where rpL5 is above the threshold for independent phase separation with NPM1. The star at 40 μM N294 indicates that the droplets experienced wetting on the slide surface and expanded above the maximum threshold of 100 μm² set in the analysis; (d) NPM1 constructs titrated into puncta formed between 50 μM rpL5 and 50 μg/mL rRNA, where rpL5 is below the threshold for independent phase separation with NPM1.

previously described putative NoLS involves two buried Trp residues which are essential for proper folding of the NBD (*Grummitt et al., 2008*). Mutations of these residues destabilized the NBD, thereby impeding interactions with nucleic acids and nucleolar localization (*Banuelos et al., 2013*; *Chiarella et al., 2013*; *Falini et al., 2006*; *Grummitt et al., 2008*). Thus, we propose that multi-modal binding by NPM1 may play an important role in the structural organization of the nucleolus and the recruitment of proteins to the nucleolus.

In our structural studies of the minimal phase separating system, N130/rpL5, we showed that within the liquid-like matrix of the droplets, with diameters on the microns to tens of microns length scale, we observed local ordering on the length scale of 5 to 12 nanometers (*Figures 3* and *7*). We propose that the intermolecular spacing indicated by SANS is dictated, at least in part, by the length of the linker connecting the two R-motifs in the divalent rpL5 peptide. Another factor in the spatial organization within these droplets is the 5-fold symmetry of the A1 binding grooves within the N130 pentamer that bind R-motifs within the rpL5 peptide (*Figure 1* and [*Mitrea et al., 2014*]). We suggest that the SANS data indicate the 'persistence length' of local structural order that, despite constant formation and dissociation of R-motif peptide-mediated cross-links, is sufficient to provide 'structure' to the liquid-like droplets. However, for several reasons, this is likely a highly simplified view of NPM1 structural organization within the GC of the nucleolus. First, NPM1's binding partners display diverse patterns of multivalent R-motifs (*Supplementary file 1*), providing many ways of forming cross-links with NPM1. Second, in comparison with N130, full length NPM1 exhibits the A3 acidic tract within the IDR, enabling additional interactions with R-motifs within proteins, and the

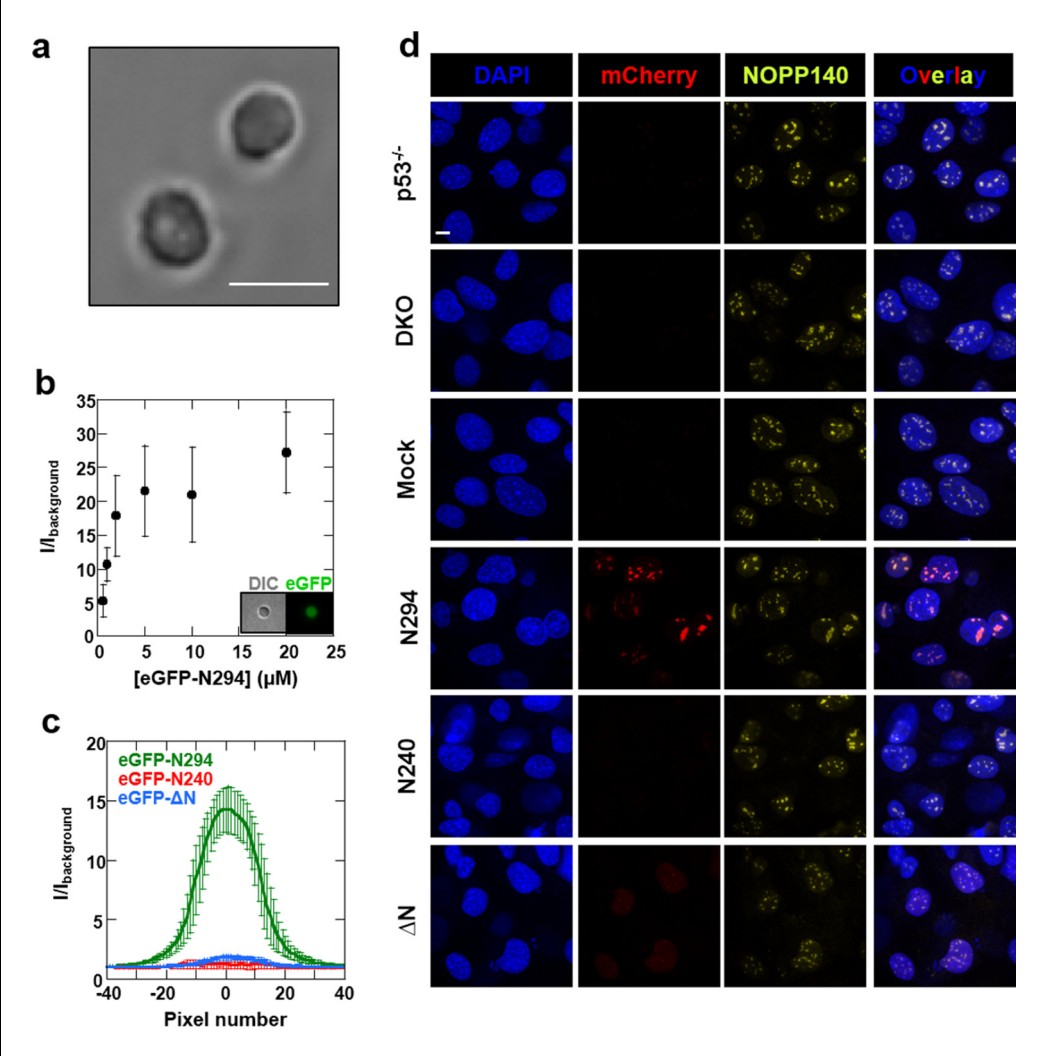

**Figure 12.** Synergistic activity of the OD and NBD are required for NPM1 incorporation in mammalian nucleoli. (a) DIC image of isolated nucleoli; (b) Titrations of recombinant eGFP-N294 into purified nucleoli. Fluorescence intensity over background is quantified. (c) Fluorescence profile through sections of nucleoli treated with 100 µM recombinant eGFP-NPM1 constructs. Values in panels (b) and (c) represent mean ± SD, n = 20 nucleoli; (d) Fluorescence confocal microscopy images of MEF cell lines; DAPI was imaged to identify nuclei, mCherry to identify the NPM1 constructs, and a fluorescently-labeled antibody to NOPP140 to identify the GC of nucleoli. Scale bar = 5µm.

The following figure supplements are available for figure 12:

**Figure supplement 1.** The multi-modal binding properties of the OD and NBD of NPM1 are required for localization within mammalian nucleoli.

**Figure supplement 2.** Flow cytometric analysis of transduced MEFs.

---

NBD, that can bind to rRNA. Therefore, due to these factors, NPM1's structural organization within the nucleolus is likely highly heterogenous, in contrast to the orderly organization of N130 within droplets with rpL5. This model for the the structurally heterogenous integration of NPM1 within the nucleolar matrix provides a mechanistic explanation for its central role in many nucleolar functions, such as ribosome biogenesis and nucleolar stress responses (*Lindstrom, 2011*).

Through studies of an *in vitro* model system based upon formation of multi-component liquid-like droplets, representing a simplified form of the GC of the nucleolus, we identified three fundamental

types of interactions that mediated phase separation: (1) R-motifs (from nucleolar proteins) binding to oligomeric, multivalent A-tracts (within NPM1), (2) low complexity multivalent R-motifs binding to rRNA and (3) oligomeric, folded nucleic acid binding domains (within NPM1) binding to rRNA. Disrupting either type (1) or (3) interactions abrogated accumulation of NPM1 within nucleoli (*Figure 12*), demonstrating the relevance of our *in vitro* findings to the physiological setting. Complete loss of NPM1 from nucleoli (e.g., in the *Trp53*[-/-]/*Npm1*[-/-] mouse embryonic fibroblasts cells) did not, however, preclude other nucleolar proteins (NOPP140 and fibrillarin; *Figure 12* and *Figure 12—figure supplement 1*, respectively) from accumulating within punctate nuclear structures, in support of a model wherein collective interactions between many different nucleolar components drive their assembly into dense liquid-like structures.

While the R-motif-rich sequences are recognized to be associated with nucleolar localization (*Scott et al., 2010*), the other two protein features identified in our study to be important for interactions with nucleolar components, namely multivalent acidic tracts and multivalent, folded nucleic acid binding domains, are not. Notably, the association of multivalency mediated by homo-oligomerization and acidic low complexity regions with phase separation is, to the best of our knowledge, unique to NPM1. We expect, however, that similar features may be utilized by other proteins for incorporation into the nucleolar matrix. For example, NOPP140, known to be localized within the GC, exhibits 11 acidic, serine-rich clusters ([*UniProt, 2014*]; http://www.uniprot.org/uniprot/Q14978). Furthermore, nucleolin, which also lacks a canonical NoLS, exhibits 3 acidic tracts (with lengths between 15 and 29 amino acids), 4 tandem folded RNA recognition motifs in addition to multivalent R-motifs ([*UniProt, 2014*]; http://www.uniprot.org/uniprot/P19338), which may mediate multi-modal binding to protein and rRNA components within the dense liquid-like phase of the nucleolus. While the established NoLS is consistent with our mechanistic model for nucleolar localization, other proteins, which we suggest function by contributing to the structural organization of the nucleolus, such as NPM1, are not encompassed by this definition. We propose that the three fundamental types of interactions noted above mediate the localization of a wide veriety of proteins, along with rRNA, within the liquid-like structure of the nucleolus, providing new perspective on the term 'nucleolar localization signal'.

## Materials and methods

### Bioinformatics analysis of NPM1 binding proteins

A list of 132 NPM1 binding partners obtained from BioGRID (deposited by Dr. Steven Gygi, Harvard Medical School; available at http://thebiogrid.org/166968/publication/high-throughput-proteomic-mapping-of-human-interaction-networks-via-affinity-purification-mass-spectrometry.html) was analyzed using the DAVID Bioinformatics Resources (http://david.abcc.ncifcrf.gov) (*Huang et al., 2009*; *2008*) to identify proteins with known involvement in nucleolar structure and/or function. GO terms were available for 125 of the 132 NPM1-binding proteins (*Supplementary file 2*). To gain insight into the molecular basis for interactions with NPM1, we determined the occurrence of multivalent R-motifs within the sequences of the 132 NPM1-binding proteins. Minimal R-motifs were defined as follows: a minimal, single R-motif as the sequence pattern, $RX_{n1}R$, where $n1 \leq 2$, R is arginine and X is any amino acid; a minimal, multivalent R-motif as the sequence pattern, $RX_{n1}RX_{n2}RX_{n3}R$, where $n1$, $n3 \leq 2$, and $n2 \leq 20$) (*Supplementary file 1*). In *Supplementary file 1*, the identified R-motifs were extended if followed by another Arg residue within two or fewer residues. A Python algorithm, available for download at https://github.com/dlaszlo88/eLIFE-NPM_NMRrelaxation was developed to identify proteins exhibiting multivalent R-motifs and was applied to the list of NPM1 binding proteins as well as a list of 20,193 non-redundant human proteins obtained from the UniProtKB/Swiss-Prot database (http://www.uniprot.org/uniprot/) (*UniProt, 2014*). Of the 132 NPM1 binding partners, 73% exhibited at least one multivalent R-motif; in comparison, only 44% of all human proteins exhibited at least one multivalent R-motif (*Supplementary file 3*). These data show NPM1 binding partners are enriched in multivalent R motifs, when compared to the majority of the human proteome ($p < 0.0001$).

## Cloning, protein expression and purification

The N130 protein was expressed and purified as described (*Mitrea et al., 2014*). The eGFP-NPM1 constructs were subcloned in the pET28 vector (Novagen, Darmstadt, Germany), with an N-terminal poly-His tag. eGFP was amplified from pEGFP-C1, a gift from Dr. Douglas Green, and inserted between NdeI and BamHI restriction sites. NPM1 construct genes, derived from human NPM1, were cloned between BamHI and XhoI restriction sites. The original Thrombin cleavage site following the affinity tag was replaced with a PreScission cleavage site. Proteins were expressed in *E. Coli* strain BL21(DE3) in Luria Broth. Protein expression was induced at $OD_{600}$ ~0.6 with 100 mg/L IPTG (Goldbio, St. Louis, MO) and the cultures were incubated overnight at 20 °C. Bacterial pellets were harvested by centrifugation and lysed by sonication in 20 mM Tris, 150 mM NaCl, 5 mM β-mercapto-ethanol (BME), pH 7.5, supplemented with protease inhibitors (SigmaFAST, Sigma-Aldrich, St. Louis, MO). eGFP-NPM1 proteins were further purified using Ni-NTA affinity chromatography, using 0.5 M NaCl in the buffers. The eluted proteins were dialysed overnight against 10 mM Tris, 50 mM NaCl, 2 mM DTT pH 7.5 in the presence of HRV3C protease (BioVision, Milpitas, CA) to remove the poly-His tag and further purified on a Mono Q HR 5/5 (GE Healthcare, Pittsburgh, PA) ion exchange column, followed by size exclusion chromatography on HiLoad 26/60 Superdex 200 (GE Healthcare, Pittsburgh, PA) in 10 mM Tris, 150 mM NaCl, 2 mM DTT, pH 7.5. The N122 construct was a gift from Dr. Yuh Min Chook (UT Southwestern, Dallas). The protein was expressed with an N-terminal GST tag as described above. The N122 protein was affinity purified on a reduced glutathione agarose column (Qiagen, Hilden, Germany), followed by GST tag proteolysis with thrombin and finally purified to homogeneity using size exclusion chromatography, as described above.

## Peptide synthesis

Peptides were synthesized in house, by the Macromolecular Synthesis resource within the Hartwell Center for Bioinformatics and Biotechnology at St. Jude Children's Research Hospital, using standard solid phase peptide synthesis chemistry.

## Phase transition assays by light scattering

The samples were incubated at room temperature for 5 min and UV-Vis absorbance spectra were recorded in triplicate on a NanoDrop 2000c spectrophotometer (Thermo Scientific, Waltham, MA).

## Fluorescent labeling

N130 was labeled with Alexa Fluor488 $C_5$ maleimide (Life Technologies, Carlsbad, CA), according to the manufacturer's protocol. The labeling was performed in 10 Tris, 150 mM NaCl, pH 7.5 buffer to maintain N130 in its folded pentameric state (*Mitrea et al., 2014*) and ensure selective labeling at solvent exposed Cys104, thereby avoiding labeling the buried Cys21. The rpL5 peptide was N-terminally labeled using Alexa Fluor594 Succinimidyl ester (Life Technologies, Carlsbad, CA) following the manufacturer's protocol.

## Isothermal titration calorimetry

Titrations were performed using a GE Auto-iTC200 instrument (Malvern, Malvern, UK), at 25 °C, with the NPM1 construct in the cell. Proteins and peptides were dialyzed overnight against the reaction buffer, consisting of 10 mM Na phosphate, 150 mM NaCl, 2 mM DTT, pH 7.0. The concentrations were selected to be below the phase separation threshold and were determined from $A_{280nm}$ of the protein or peptide diluted in 6 M guanidine hydrochloride buffer.

## Fluorescence spectroscopy

N130 constructs with Q15C and S125C single and double mutations in the background of a C104T mutation were used. For ensemble fluorescence anisotropy measurements, N130 labeled at C125 with Alexa Fluor594 ($C_5$ maleimide derivative; Life Technologies, Carlsbad, CA) was used. The protein was labeled in 10 mM Na phosphate, 500 mM NaCl, pH 7.5 with 2-fold molar excess of the dye overnight at 4 °C in the dark. For dual labeling of N130 at C15 and C125, the following protocol was used: a 2-fold molar excess of donor dye (Alexa Fluor594 C5-maleimide derivative, Life Technologies, Carlsbad, CA), and a 8-fold molar excess of the acceptor dye (Alexa Fluor680 C2-maleimide derivative, Life Technologies, Carlsbad, CA) were incubated together in 10 mM Na

phosphate, 500 mM NaCl, pH 7.5 overnight at 4 °C for in the dark. The excess dye in all labeling reactions was removed by multiple rounds of washing with the labeling buffers using a 3K MWCO centrifugal filter device (Millipore, Darmstadt, Germany). The purity of all the samples was confirmed by ESI-mass spectroscopy (Scripps Center for Mass Spectrometry). Ensemble fluorescence measurements were carried out using an automated temperature controlled PC1 spectrofluorometer (ISS, Champaign, IL) in 10 mM Tris, 150 mM NaCl, pH 7.5. The single-molecule fluorescence measurements were performed on freely diffusing molecules with a 532 nm excitation line (CrystaLaser, Reno, NV) (operating on a 500 µW power) using a home-built instrument. The details of smFRET instrumentation, data collection and data analysis have been described elsewhere (*Ferreon et al., 2009*; *2013*). All experiments were performed at room temperature in 10 mM Tris, 150 mM NaCl, 2 mM DTT, pH 7.5 using a dual-labeled protein concentration of ~100 pM in the presence of 200 µM unlabeled N130. To minimize photobleaching, 0.5% N-propyl gallate (final concentration) was used from a 200x concentrated stock in acetonitrile.

## NMR spectroscopy

For all NMR experiments, the N130 construct was used. Samples of perdeuterated, $^{15}N$ uniformly labeled, N130 were dissolved in 90% $H_2O$/10% $D_2O$ containing 10 mM sodium phosphate buffer at pH 7.0 with 150 mM NaCl and 5 mM DTT. All experiments were performed at 298 K. TROSY-HSQC based titration experiments were collected with a Bruker Avance I spectrometer operating at a Larmor frequency of 800 MHz (Bruker, Billerica, MA). $^{15}N$ backbone relaxation experiments which measured the transverse cross-relaxation and longitudinal relaxation rates were performed on the free and bound N130 at static magnetic fields of 600, 800 and 1000 MHz. For N130 in the droplet state the same experiments were performed, but only at 600 and 800 MHz. Data were processed and visualized using NMRPipe and CARA, respectively. Fits to all NMR data for the extraction of parameters describing the observed motion were performed using in-house written scripts in Python and Mathematica (Wolfram Research, Champaign, IL). All scripts are available for download at available for download at https://github.com/dlaszlo88/eLIFE-NPM_NMRrelaxation. See below for details on all NMR experiments and analysis.

## Analytical ultracentrifugation

Sedimentation velocity experiments were conducted in a ProteomeLab XL-I analytical ultracentrifuge (Beckman Coulter, Indianapolis, IN) following standard protocols unless mentioned otherwise (*Schuck, 2000*; *Zhao et al., 2013*). The samples in a buffer containing 10 mM NaP pH 7.0, 150 mM NaCl, and 2 mM DTT were loaded into a cell assembly comprised of a double sector charcoal-filled centerpiece with a 12 mm path length and sapphire windows. The cell assembly, containing identical sample and reference buffer volumes of 380 µL, was placed in a rotor and temperature equilibrated at rest at 20 °C for 2 hr before it was accelerated from 0 to 40000 rpm. Rayleigh interference optical data were collected continuously for 12 hr, 400 min for 1:0 and 1:1 and 300 min for 1:2 and 1:3 and analyzed. The velocity data were modeled with diffusion-deconvoluted sedimentation coefficient distributions c(s) in SEDFIT (https://sedfitsedphat.nibib.nih.gov/software/default.aspx), using algebraic noise decomposition and with signal-average frictional ratio and meniscus position refined with nonlinear regression. The s-value was corrected for time, temperature and radial position and finite acceleration of the rotor was accounted for in the evaluation of Lamm equation solutions (*Zhao et al., 2015*). Maximum entropy regularization was applied at a confidence level of P-0.68.

## Small-angle neutron scattering

SANS experiments were performed on the extended Q-range small-angle neutron scattering (EQ-SANS, BL-6) beam line at the Spallation Neutron Source (SNS) located at Oak Ridge National Laboratory (ORNL). In 30 Hz operation mode, a 4 m sample-to-detector distance with 2.5–6.1 and 9.4–13.4 Å wavelength bands was used (*Zhao et al., 2010*) to obtain the relevant wavevector transfer, $q = 4\pi \sin(\theta)/\lambda$, where $2\theta$ is the scattering angle. rpL5:N130 samples at 0:1, 1:1, 2:1, and 3:1 mol ratios were prepared in 10 mM Tris, 150 mM NaCl, 2 mM DTT $D_2O$ (pH measured = 7.5). The samples were loaded into 2 mm pathlength circular-shaped quartz cuvettes (Hellma USA, Plainville, NY) and SANS measurements were performed at 25 °C. Data reduction followed standard procedures using MantidPlot (*Arnold et al., 2014*). The measured scattering intensity was corrected for the detector

sensitivity and scattering contribution from the solvent and empty cells, and then placed on absolute scale using a calibrated standard (*Wignall and Bates, 1987*).

## Light microscopy of in vitrodroplets

Wide field DIC and 488 nm fluorescence images used for phase separation assays with the N130 construct were collected on a Nikon C1Si microscope (Nikon Instruments, Melville, NY) using a 60x 1.45 NA magnification oil objective. 10 μl samples were incubated at room temperature for 5 min prior to analysis. Droplets were defined as having an area greater than 9 squared pixels (0.2 μm/pixel) and circularity 0.5–1. Particles smaller than 9 squared pixels were visible by fluorescence but not DIC at low protein and peptide concentrations; however, these objects were below the threshold to confidently measure circularity. Therefore, droplets formation was not recorded until particles could be observed by DIC.

For ternary mixture assays, the various AlexaFluor488-tagged NPM1 constructs were titrated into the mixtures of rRNA and rpL5 peptide. Confocal images of phase separation droplets were collected on Nikon C1Si (Nikon Instruments, Melville, NY) or Zeiss Axio Observer (Carl Zeiss Microscopy, Jena, Germany) microscopes, using a 60X 1.45 NA or 63X 1.40 NA oil magnification objective, respectively, in μ-slide VI$^{0.4}$ 6 channel flow cells (ibidi, Madison, WI) or CultureWell 16-well chambered coverglass (Grace Biolabs, Bend, OR), coated with PlusOne Repel-Silane ES (GE Healthcare, Pittsburgh, PA). Images of MEFs and isolated nucleoli were recorded on a Zeiss Axio ObserverZ.1 microscope equipped with a CSU-22 spinning disk (Yokagawa, Tokyo, Japan), Delta Evolve EMCCD camera and 100X 1.45 NA oil objective.

## Isolation of nucleoli

Nucleoli were prepared from Cos7 cells (ATCC number CRL 1651) using an established protocol (*Rosner et al., 2013*). Briefly, $1 \times 10^7$ cells were harvested by trypsin digestion followed by serum inactivation and washing in two exchanges of PBS. Cells were incubated in nucleolar isolation buffer (NIB; 10 mM Tris, 2 mM MgCl$_2$, 0.5 mM EDTA, pH 7.5) for 2 min at room temperature followed by 10 min on ice. The cells were disrupted by the addition of NP-40 to a final concentration of 1%. The mixture was centrifuged for 3 min at 500 x g to collect the nuclear fraction, followed by washing in NIB containing NP-40 prior to resuspension in buffer without additional NP-40. The nuclear suspension was sonicated on ice at 20% power for 12 cycles of 1 s ON / 5 s OFF, using a 1/8'' tip on a Fisher Scientific Model 505 Sonic Dismembrator (Thermo Fisher Scientific, Waltham, MA). Nucleoli were collected as the pellet by centrifugation at 4 °C, 500 x g, for 3 min and resuspended in NIB. The integrity of purified nucleoli was verified by (i) morphology and (ii) distribution of established nucleolar proteins via immunostaining and confocal microscopy. We note that incubation in the presence of 150 mM NaCl to match the physiological millieu caused isolated nucleoli to dissolve.

## Cell culture

*Trp53*$^{-/-}$ and *NPM1*$^{-/-}$ / *Trp53*$^{-/-}$ (DKO) MEF cell lines were a kind gift from Drs. C. Sherr (St. Jude Children's Research Hospital) and P. Pandolfi (Beth Israel Deaconess Medical Center). Retroviruses were produced by transfecting the Thy1.1-IRES-mCherry-NPM1 variant plasmids or the empty vector into Phoenix cells, using Xfect transfection reagent (Clontech Laboratories, Mountain VIew, CA), in 6 well plates. We note that all Phoenix cells harboring plasmids that encoded for mCherry exhibited red fluorescence under the light microscope. Transfected cells were incubated overnight at 37 °C in a 5% CO$_2$ atmosphere. The supernatant containing viruses was collected, filtered through a 0.45 μm seringe filter, and transferred immediately onto 6 well plates seeded with $8 \times 10^4$ DKO MEFs, in the presence of 5 μg/mL polybrene. Infections were carried twice a day, for a total of five infections. Virally transduced cells were sorted based on Thy1.1 expression using FACS; sorted cells were expanded in culture.

## Flow Cytometry

MEFs were trypsinized using 0.05% trypsin + EDTA prior to resuspension in PBS containing 1% BSA and 1 mM EDTA. Cells were surface labeled with APC-conjugated anti-Thy1.1 (clone OX-7; Biolegend, San Diego, CA) at [0.4 ng/ml] for 30 min on ice. Following incubation, cells were washed three

times prior to resuspension in PBS containing 1 mM EDTA and analysis using a Fortessa cytometer (BD Biosciences, San Jose, CA).

## Image processing

Enrichment of eGFP-tagged NPM1 constructs in purified nucleoli was determined using Slidebook 6.0 (Intelligent Imaging Innovations, Gottingen, Germany). Briefly, each nucleolus was bisected with a line of 80 pixels in length, based on the DIC image. The resulting fluorescence intensity of each pixel along the line was plotted according to pixel position.

For the quantification of droplet size, the lower and upper limits of fluorescence intensity were matched for all images in Slidebook 6.0 and the particle count was subsequently performed in ImageJ, as follows: 24-bit RGB images were converted into 8-bit images, background subtraction and auto threshold were applied, images were converted to mask and merged objects were separated using the binary watershed operation. Particles were analyzed using Analyze Particles, imposing the size restriction of 5–2500 pixel (~100 µm$^2$) and the circularity restriction 0.2–1. >90% of all observed droplets had areas within the selected window and were included in the analysis, unless otherwise noted; >50 particles were counted per image.

## Immunostaining

Purified nucleoli were allowed to settle onto poly-D-Lysine-coated chambered coverslips prior to fixation for 10 min with 4% paraformaldehyde in PBS. Nucleoli were subsequently treated for 10 min with cold acetone at -20 °C prior to incubation with blocking buffer (20 mM Tris pH 8.0, 100 mM NaCl, 2% bovine serum albumin, 0.05% Tween-20) for 30 min. The samples were stained overnight at 4 °C in blocking buffer containing anti-Fibrillarin (Novus Biologicals, Littleton, CO; NB300-269, 2.5 µg/mL), anti-rpL5 (Santa Cruz Biotech, Dallas, TX; sc-103865, 2 µg/mL) and anti-NPM (Abgent, San Diego, CA; AP2834b, 1:200 dilution) antibodies. The samples were washed in 20 mM Tris pH8.0, 100 mM NaCl and 0.05% Tween-20 prior to detection with fluorescently labeled secondary antibodies (Life Technologies, Carlsbad, CA) and imaged using a Zeiss Axio Observer Z.1 equipped with a CSU-22 spinning disk, Delta evolve EMCCD camera and 100X 1.45NA oil objective, and Slidebook 6.0 (Intelligent Imaging Innovations, Gottingen, Germany).

MEFs were seeded onto 4-well chambers at 40000 cells/well and incubated overnight at 37 °C, in 5% $CO_2$ incubator. Cultured MEFs were fixed in 4% methanol-free paraformaldehyde for 10 min at room temperature, followed by incubation in TBS (20 mM Tris, 100 mM NaCl) containing 0.3 M glycine for 10 min. The cells were permeabilized for 3 min in TBS containing 0.1% Triton-100, followed by 30 min incubation in TBS containing 2% bovine serum albumin. The slides were incubated overnight at 4 ° with anti-Fibrillarin (clone 38F3; Genetex, Irvine, CA) or anti-Nopp140 (sc-28672, Santa Cruz Biotech, Dallas, TX) antibodies each at [1 ng/ml] diluted in TBS + 2% BSA. Primary antibodies were detected with either AF647-labeled goat anti-Mouse (ThermoFisher) or AF647-labeled goat anti-Rabbit (ThermoFisher) secondary antibodies at [1 ug/ml] for 1 hr at room temperature. Samples were post-fixed in 1% PFA after washing, and prior to imaging in TBS containing 1 ug/ml Hoescht (Thermo Fisher Scientific, Walthman, MA). Images were acquired using a Marianas spinning-disk laser scanning confocal (Intelligent Imaging Innovations) comprising a Zeiss AxioObserverZ.1 equipped with a 63x 1.4NA objective and Evolve EMCCD camera (Photometrics, Tucson, AZ).

## Analysis methods
### SANS data analysis

#### 0:1 and 1:1 rpL5:N130

Upon verifying a Guinier regime (*Guinier and Fournet, 1955*) in the rpL5:N130 0:1 and 1:1 SANS profiles, the pair distance distribution function, *P(r)*, was calculated from the scattering intensity, *I(q)*, using the indirect Fourier transform method implemented in the GNOM program (*Svergun, 1992*) (*Figure 3b*). The *P(r)* function was set to zero for *r* = 0 and *r* = $D_{max}$, the maximum linear dimension of the scattering object. $D_{max}$ was explored to optimize the *P(r)* solution and excellent quality solutions were found in each case. The real-space radius of gyration, $R_g$, and scattering intensity at zero angle, *I(0)*, were determined from the *P(r)* solution to the scattering data. The molecular mass, *M*, was calculated by

$$I(0) = \frac{cM}{N_A}(\Delta\rho)^2 \overline{\vartheta} \qquad (1)$$

where $c$ = protein concentration (= 3 mg/mL), $\Delta\rho$ = contrast in scattering length density between protein and $D_2O$ buffer solution (= $\rho_{prot} - \rho_{D2O}$), $\overline{\vartheta}$ = protein partial specific volume (= 0.73 ml/g), and $N_A$ = Avogadro's number. The N130 scattering length density, $\rho_{prot}$, (= $3.0 \times 10^{10}$ cm$^{-2}$) was calculated from the sequence using CRYSON (*Svergun et al., 1995*). The $D_2O$ scattering length density used was $\rho_{D2O} = 6.404 \times 10^{10}$ cm$^{-2}$.

### 2:1 rpL5:N130

The rpL5:N130 2:1 SANS curve was fit to an empirical broad peak model implemented in the NCNR SANS package (*Kline, 2006*) for Igor Pro software where the scattering intensity is:

$$I(q) = \frac{A}{q^n} + \frac{C}{(q - q_0\xi)^m} + B \qquad (2)$$

to yield the Porod scale, $A$ (= $(3.1 \pm 0.3) \times 10^{-5}$), and exponent, $n$ (= $2.68 \pm 0.02$), Lorentzian scale, $C$ (= $0.0394 \pm 0.0007$), peak position, $q_0$ (= $0.0740 \pm 0.0004$ Å$^{-1}$), screening length, $\xi$ (= $36.7 \pm 0.5$ Å), and exponent, $m$ (= $3.46 \pm 0.07$), and background, $B$ (= $(-3 \pm 2) \times 10^{-4}$ cm$^{-1}$). The peak position corresponds to a $d$-spacing ($d_0 = 2\pi/q_0$) of 85 Å.

### 3:1 rpL5:N130

A portion of the rpL5:N130 3:1 SANS curve was fit to three Gaussian peaks using the multi-peak fitting routine within Igor Pro software and yielded peak maxima at $q_1 = 0.0528$ Å$^{-1}$, $q_2 = 0.0820$ Å$^{-1}$, and $q_3 = 0.1140$ Å$^{-1}$ with corresponding $d$-spacings ($d_i = 2\pi/q_i$) of $d_1 = 119$ Å, $d_2 = 77$ Å, and $d_3 = 55$ Å, respectively.

## NMR analysis of $^2$H/$^{15}$N-N130; apo, bound to rpL5 in the solution phase, and bound to rpL5 in liquid-like droplets

### Characterization of rpL5 binding to $^2$H/$^{15}$N-N130 in the solution phase state; NMR-based titration

N130 for all NMR samples was maintained at a monomer concentration of 284 µM. NMR-based chemical shift titrations of N130 with rpL5 peptide were performed in a step-wise manner in which a stock of highly concentrated peptide was added incrementally to the N130 samples. A total of nine additions were performed over which the dilution of the N130 was small changing the overall volume of the sample by only 2%. rpL5 was titrated into a sample with N130 up to a [rpL5]:[N130] ratio of 3:1 (maximum concentration of rpL5 was 870 µM). All titration experiments were collected with a Bruker Avance I spectrometer operating at a Larmor frequency of 800 MHz. For apo N130 and with each addition of rpL5 a TROSY type amide proton, nitrogen two-dimensional correlation spectrum was recorded. Spectra were collected with 100 and 1024 complex points in the indirect and direct dimension yielding maximum acquisition times of 12.1 ms and 91.8 ms, respectively. Incremental additions of rpL5 allowed for the extraction of binding isotherms by monitoring the change in the $^{15}$N chemical shift ($\Delta\omega_{15N}$) as a function of ligand concentration (*Table 4*). Binding isotherms were fit to a simple two-state binding model and used to extract the dissociation constant ($K_D$) of the interaction (*Cavanagh et al., 2007*). Although N130 displayed a range of site-specific $K_D$ values (*Table 4*), we established criteria for the selection of isotherms that were considered for the global determination of the $K_D$, as follows.

First, only nuclei whose $\Delta\omega_{15N}$ exceeded 0.15 ppm by the end of the titration were considered. This yielded a total of 36 individual binding isotherms. Second, for each binding isotherm, the last five points were normalized with respect to their final observed $\Delta\omega_{15N}$ during the titration ($\Delta\omega_{15N}$ at a [rpL5]:[N130] at 3:1) and fit to a straight line. Slopes from these fits that were close to zero indicated flat binding isotherms at the end of the titration and $^{15}$N nuclei that had reached saturation (e. g., their chemical shifts do not change significantly at higher [rpL5]:[N130] ratios). $^{15}$N nuclei whose slope was less than the average were pooled and used for a global analysis of the $K_D$ value. This provided a total of 18 binding isotherms which were fit to a single $K_D$ value and to each binding isotherm's individual maximum $\Delta\omega_{15N}$. The global analysis was implemented in Mathematica (Wolfram

Research, Champaign, IL) and yielded a global $K_D$ value of 57 ± 14 µM and is similar to the $K_D$ value determined using isothermal titration calorimetry (21 ± 5 µM; *Table 2*). We appreciate that for a given pentamer each monomeric unit consists of two possible binding sites (A1 & A2 tracts; *Figure 1a*) and the current use of a simple two-state binding model does not distinguish between these two sites. The extension to higher order binding models that represent multiple independent binding sites may be necessary, but the binding isotherms collected here do not display multiple independent binding events and thus only a two-state model was applied (*Arai et al., 2012*). Error bars for all individual and global fits were determined from Monte-Carlo simulations in which a 10% error in the absolute peptide concentration (effectively shifting points along the abscissa) was considered and 500 random binding isotherms were generated and subsequently refit (*Markin and Spyracopoulos, 2012*). The standard deviation in parameters from these simulations was used as the error for these measurements.

## Method for analysis of fast pico- to nanosecond motions in apo N130 and solution phase rpL5:N130 complexes

We sought quantitative, atomic resolution information on the dynamic features of $^2$H/$^{15}$N-N130 in three different states: apo form, bound to rpL5 in the solution phase, and bound to rpL5 in liquid-like droplets. Since solution NMR is uniquely suited to probe motions with atomic detail, we exploited a combination of NMR relaxation experiments that report on motions faster than the overall rotational correlation time ($\tau_c$). Conventional NMR relaxation experiments which probe fast pico- to nanosecond timescale motion can be used to obtain information on internuclear vector flexibility ($S^2$) and molecular time constants related to rotational diffusion processes (*Palmer, 2004*). These experiments, when combined, can be used to extract $\tau_c$, $S^2$, and time constants such as the fast lifetimes of internal motion ($\tau_f$; picoseconds), and in some cases motion in the low nanosecond range if $\tau_c$ is much greater than this additionally 'slow' nanosecond motion (denoted as $\tau_s$). However, for high molecular weight proteins like N130 (the molecular weight of the pentamer is 73.4 kDa), the study of backbone $^{15}$N nuclei dynamics is challenging due to slow molecular tumbling. In order to overcome this, we utilized experiments analogous to Lakomek, *et al.*, (*Lakomek et al., 2013*) that utilize the TRACT (*Lee et al., 2006*) concept in which the slowly (β) and fast relaxing (α) transverse components of the $^{15}$N doublet were individually queried and their rate of decay measured within conventional TROSY-type HSQC experiments in order to obtain the relaxation rates for both components. Measurement of both $R_{2,\beta}$ and $R_{2,\alpha}$ permits the calculation of the transverse cross-relaxation rate, $\eta_{xy}$, which reports on motions up to $\tau_c$ and in addition contains no contribution of slow ms-µs motional processes due to chemical exchange (*Lee et al., 2006*). For the extraction of parameters that define the observed relaxation processes, we also recorded longitudinal relaxation ($R_1$) experiments for backbone $^{15}$N nuclei which have small values for large molecular weight proteins to increase the number of observables (*Lee and Wand, 1999*). The inclusion of $R_1$ values in our analysis was critical because this relaxation rate samples contributions of high frequency motions for backbone $^{15}$N nuclei that $\eta_{xy}$ does not (*Kroenke et al., 1998*). The approach described above was necessary because high quality intrinsic relaxation rate ($R_{2,0}$) and $^{15}$N heteronuclear-NOE data could not be recorded due to signal-to-noise limitations.

## NMR relaxation data measurements

The samples of apo N130 were maintained at a monomer concentration of 215 µM and measurements on solution phase, rpL5-bound N130 contained 190 µM N130 and 475 µM rpL5 (2.5:1 rpL5: N130). $R_{2,\alpha}$ and $R_{2,\beta}$ experiments were performed at three static magnetic fields with Larmor frequencies of 600 (Bruker Avance III), 800 (Bruker Avance I), and 1000 (Bruker Avance III) MHz. This was necessary to have enough independent observables to uniquely determine the parameters that define the relaxation rates. Since N130 consists of a folded pentameric core and disordered N- and C-termini, proper sampling of the relaxation profiles was necessary. Therefore, experiments were recorded with different delay times for resonances from the folded core and the disordered N- and C-termini. For $R_{2,\alpha/\beta}$, measurements at 800 and 1000 MHz were subjected to full relaxation delay (*Figure 4b*,i-ii). All experiments were executed by interleaving the employed delay times. At 600 MHz, $R_{2,\alpha/\beta}$ rates were measured using a two-point sampling scheme (*Jones et al., 1996*) at the estimated average value of $1/R_{2,\alpha/\beta}$.

Typical acquisition parameters for the $R_{2,\beta}$ experiment employed 90 and 1024 complex points in the indirect and direct dimension, respectively. For each point, 128 transients were measured and the total experiment time was 32 hr. For $R_{2,\alpha}$, the same number of points in both indirect and direct dimensions were taken, but the number of transients per point was increased to 160. A recycle delay of one second was used for all experiments. Additional $R_{2,\alpha/\beta}$ experiments were also performed with narrowed sweep widths that focused on the residues from the disordered N- and C-termini with 70 and 1024 complex and 16 transients per point increment. We implemented the $R_1$ experiment with TROSY-readout similar to that in Lakomek, *et al.* (*Lakomek et al., 2012*), and performed experiments in which 70 and 512 complex points in the indirect and direct dimension, respectively, were acquired with 120 transients per point (*Figure 4b*,iii). The recycle delay for the $R_1$ experiments was set to two seconds yielding a total experiment run time of 38 hr. All sampled delay lengths were interleaved in a random order throughout the experiment.

## Quantification of NMR relaxation rates

All spectra were processed with the NMRPipe package (*Delaglio et al., 1995*), visualized with CARA, and analyzed with in-house scripts written in Python using the Scipy computing libraries. Intensity profiles as a function of delay time (*Figure 4b*,i–iii) were fitted to an exponential decay function in which a single relaxation rate and initial intensity at time zero were extracted. Errors in the measured relaxation rates were determined from Monte-Carlo sampling with 1000 runs where either the base-plane noise or duplicate points were used for the standard deviation in the assumed normal distribution and whose average was placed at the measured intensity value. Errors in $R_{2,\alpha/\beta}$ data recorded at 600 MHz were propagated from the base-plane noise in each spectrum. From the $R_{2,\alpha}$ and $R_{2,\beta}$ measurements, $\eta_{xy}$ values were calculated as half their difference and the error in $\eta_{xy}$ was determined by error propagation.

A total of 477 (84) individual $R_{2,\alpha/\beta}$ ($R_1$) relaxation rates were quantified for apo N130, and for solution phase, 2.5:1 rpL5:N130, a total of 402 (74) $R_{2,\alpha/\beta}$ ($R_1$) relaxation rates were collected. From the $R_{2,\alpha/\beta}$ data, a total of 212 and 183 $\eta_{xy}$ values across all static magnetic fields employed here could be calculated for apo N130 and 2.5:1 rpL5:N130, respectively. We note that the static magnetic field dependence of the absolute value of $R_1$ for a protein as large as apo N130 or 2.5:1 rpL5:N130 is small. Although the expected relative difference in $R_1$ from 600 to 800 MHz is 32%, and from 800 MHz to 1000 MHz is 22%, the folded core of N130 can be expected to have values between ~0.1 and 0.3 s$^{-1}$. Since the average precision in $R_1$ for the free N130 ($R_1^{all} = 0.47 \pm 0.07$ s$^{-1}$; $R_1^{folded-core} = 0.32 \pm 0.08$ s$^{-1}$) and 2.5:1 rpL5:N130 ($R_1^{all} = 0.44 \pm 0.10$ s$^{-1}$; $R_1^{folded-core} = 0.28 \pm 0.11$ s$^{-1}$) was 15% and 23% of the average measured rate, respectively, at this level of precision, only $R_1$ data at 800 MHz was utilized in the calculations described below. Ultimately, a total of 296 and 257 independent values could be used for further calculation yielding 60% and 51% coverage of the backbone for apo N130 and 2.5:1 rpL5:N130, respectively, yielding approximately four observables per $^{15}$N nucleus. For the 2.5:1 rpL5:N130 sample, which experiences exchange with peptide in the fast regime, the $\eta_{xy}$ data represent the population weighted average between the free and the bound states. Regardless, we used the measured $\eta_{xy}$ values of 2.5:1 rpL5:N130 because N130 was extensively saturated with the rpL5 peptide (93% bound). Bound rates from the population weighted measured rates were calculated, but were found to be within error, and did not deviate from the observed values because of high degree of N130 saturation with the rpL5 peptide.

## Determination of $\tau_c$ for apo N130 and 2.5:1 rpL5:N130

$\eta_{xy}$ and $R_1$ are composed of a sum of spectral density functions evaluated at particular frequencies $(J(\omega))$ which maintain a Lorentzian shape and particular prefactors whose explicit form have been reported previously (*Abragam, 1961*; *Cavanagh et al., 2007*; *Palmer, 2004*). In all calculations, $^{15}$N CSA was set to -162 ppm (*Yao et al., 2010a*), the $^{15}$N-$^1$H$^N$ bond length was 1.04 Å (*Yao et al., 2008*), and the angle between CSA tensor and the unique axis of the N-H bond (*Lee et al., 2006*) was set to 17°. All other constants that compose the prefactors for $\eta_{xy}$ and $R_1$ were set to their known values (*Cavanagh et al., 2007*). We utilized the Lipari-Szabo Model-Free formalism (*Lipari and Szabo, 1982*) (LS-MF) to describe the spectral density functions $J(\omega) = \frac{2}{5}\left(\frac{S^2\tau_C}{1+(\varpi\tau_C)^2} + \frac{(1-S^2)\tau_i}{1+(\varpi\tau_i)^2}\right); \tau_i = \left(\frac{1}{\tau_C} + \frac{1}{\tau_f}\right)$ in which $S^2$ and $\tau_f$ correspond to the $^1$H$^{N-15}$N internuclear vector

flexibility (values less than one indicate increased flexibility) and fast picosecond internal motion for a given internuclear vector, respectively (*Lipari and Szabo, 1982*). Using these expressions that describe the relaxation rates of each nucleus, residues were initially subjected to a grid search. This initial search consisted of only residues from the folded core of N130 which were identified based on the crystal structure (PDB: 4N8M). This was done to ensure that they would behave as resonances whose expected $^1H^{N-15}N$ NOE would be greater than 0.65 following conventional protocols for establishing residues which can be used to determine $\tau_c$ (*Kay et al., 1989*). The following method is similar to Lakomek, *et al.* (*Lakomek et al., 2013*), and Lorieau, *et al.* (*Lorieau et al., 2011*). The grid was evaluated from a range of 0.6 to 1.0 in steps of 0.01 for $S^2$ and 0 to 1 ns in steps of 100 ps for $\tau_f$. $\tau_c$ was evaluated from 40 to 80 ns in steps of 1 ns. A target function ($\chi^2$) comprised of the sum of the error weighted squared difference between the experimental and calculated relaxation rates was minimized. All lowest $\chi^2$ values from each residue's minimization was kept and nuclei with similar $\tau_c$ values and whose $\tau_f$ values were either 0 or 100 ps were grouped together and then globally minimized. This global minimization was not conducted in a grid fashion and the minimized parameters included a single $\tau_c$ and residue specific $S^2$ and $\tau_f$ parameters. The lower limit in $\tau_c$ was constrained based on the $\eta_{xy}$ data assuming a rigid rotor spectral density function (*Lee et al., 2006*) and the error in $\tau_c$ was determined from Monte-Carlo simulations with all calculations performed in Mathematica (Wolfram Research, Champaign, IL). Rotational anisotropy was not considered under the current analysis; however, the disordered N- and C-termini do fluctuate independently from the folded core based on 1D-TRACT (*Lee et al., 2006*) data in which two overall correlation times could be extracted (data not shown).

## Determination of binding stoichiometry for 2.5:1 rpL5:N130

The $\tau_c$ values discussed above were used as a quantitative assessment of the binding stoichiometry for NPM. $\tau_c^{apoN130}$ and $\tau_c^{2.5:1rpL5:N130}$ were found to be 53.6 ± 0.9 and 67.8 ± 1.1 ns, respectively. $\tau_c^{2.5:1rpL5:N130}$ was 27 ± 4% larger than that of $\tau_c^{apoN130}$. Since $\tau_c$ can be a direct reporter in the change of molecular weight ($M_r$), we calculated the theoretical $\tau_c$ value for apo N130 ($M_r$ = 73.4 kDa) and 2.5:1 rpL5:N130 using the Stokes-Einstein equation ($\tau_{c,Stokes-Einstein}$) (*Cavanagh et al., 2007*). For 2.5:1 rpL5:N130, we postulated that for each monomeric unit of the pentamer two peptides would bind, one to A1 and A2, yielding a molecular weight of 95.6 kDa. The mass increase in the 2.5:1 rpL5:N130 complex reflects the change in radius of hydration ($r_H = \left(\frac{3VM_r}{4\pi N_a}\right)^{\frac{1}{3}} + r_w$) due to peptide binding, in which V, $N_a$, and $r_w$ are the specific volume of the protein, Avogadro's number, and radius of water ($r_w$ = 2 Å), respectively. Formally, $\tau_{c,Stokes-Einstein}^{2.5:1rpL5:N130}/\tau_{c,Stokes-Einstein}^{apoN130} =$

$$\frac{\left(2r_w+\left(\frac{6}{\pi}\right)^{\frac{1}{3}}\left(\frac{VM_{r,95.6kDa}}{N_a}\right)^{\frac{1}{3}}\right)^3}{\left(2r_w+\left(\frac{6}{\pi}\right)^{\frac{1}{3}}\left(\frac{VM_{r,73.4kDa}}{N_a}\right)^{\frac{1}{3}}\right)^3}$$ and upon evaluation an excellent agreement between the experimentally determined ratio ($\tau_c^{2.5:1rpL5:N130}/\tau_c^{apoN130}$) and the ratio calculated theoretically using the relation above was found. Namely, that $\tau_{c,Stokes-Einstein}^{2.5:1rpL5:N130}$ is 28% larger than that of $\tau_{c,Stokes-Einstein}^{apoN130}$ further confirming a 2:1 (rpL5:N130$^{monomer}$) binding stoichiometry for each monomeric unit within the N130 pentamer. We also further validated the experimentally determined free N130 $\tau_c$ value using hydrodynamic calculations with the software HYDRONMR (*Ortega and Garcia de la Torre, 2005*) with structures of N130 that had its N- and C-termini modeled into the structure. The calculated value from HYDRONMR was 56.3 ns and is within 5% of the experimentally determined value.

## Insights into ps to ns motions of the N- and C-termini for apo N130 and 2.5:1 rpL5:N130

Following determination of global $\tau_c$ values, local parameters for each residue were evaluated. Either the LS-MF formalism was used or a local MF (local-MF) formalism in which residues were treated completely locally using a single $\tau_c$ value ($\tau_{c,local}$) and $S^2$. Local-MF is one in which $\tau_f$ in LS-MF equals zero. The selection of formalisms was based on fit statistics comparing models with different numbers of parameters with an F-test at 95% confidence interval and on whether minimizations using a

particular formalism produced unrealistic values (e.g. $S^2 \geq 1$) (*Lee et al., 1997*). For residues Asp3 to Asp122, the LS-MF formalism was used with the globally determined $\tau_c$ for the apo and rpL5-bound forms held constant with $S^2$ and $\tau_f$ determined. For residues in the C-terminal A2 tract consisting of Ala123-Glu130, only $\tau_{c,local}$ and $S^2$ values were determined (local-MF) (*Table 5*). Back calculation of all data from the determined parameters yielded an excellent correlation with experimental data. The Pearson correlation coefficient between $\eta_{xy}$ calculated and observed for all data across all static magnetic fields used here was 0.97 and 0.94 for the apo N130 and 2.5:1 rpL5:N130, respectively. The Pearson correlation coefficient for the $R_1$ data collected for apo and rpL5-bound N130 was 0.99 for both datasets. All errors were assessed using Monte-Carlo simulations. This procedure was performed for all data on apo N130 and 2.5:1 rpL5:N130. We note that there was a distinct lack of precision in determining $\tau_f$ for residues in which LS-MF was used to analyze the data ($\tau_f^{N130} = 212 \pm 272ps, \tau_f^{rpL5:N130} = 272 \pm 310ps$). This is expected because high frequency motions contribute to a greater extent to $R_1$ than to $\eta_{xy}$ and, with limited $R_1$ values only measured at a single magnetic field and for resonances whose $R_1$ is larger than the average, error will cause a decrease in precision when determining $\tau_f$. We note that the average $S^2$ for both apo and rpL5-bound forms of N130 are comparable to what is expected for folded proteins(*Lakomek et al., 2013*) ($S^2_{apoN130} = 0.73 \pm 0.06$ and $S^2_{2.5:1rpL5:N130} = 0.71 \pm 0.10$; average included residues 16–32 and 40–118). Of interest are significant changes to dynamic parameters for residues within the N- and C-termini of N130 due to rpL5 peptide binding (*Figure 4c*,i–iii).

Although the N- and C-termini remain disordered, the N-terminus exhibits increased rigidity as indicated by the changes in $S^2$ values and whose relaxation behavior was best described using the globally determined $\tau_c$ that was fixed during the LS-MF analysis (*Figure 4c*,i). Overall, internuclear vector motions were reduced by two-thirds with binding of rpL5 to N130 ($S^2_{apoN130} = 0.16 \pm 0.01 versus S^2_{2.5:1rpL5:N130} = 0.26 \pm 0.02$). The extraction of parameters that define C-terminal A2 tract motion was not well fit when the overall $\tau_c$ was fixed, as in the aforementioned analysis. Instead, for the C-terminal A2 tract (*Figure 4c*,ii-iii), changes in motion were monitored using $S^2$ and the overall local correlation time ($\tau_{c,local}$). For the A2 tract, this is consistent with slowed local motion and increased rigidity due to peptide binding (green bars; *Figure 4c*,ii). We note that increased rigidity can be seen beginning from residue Val119. However, we focus on the local-MF analysis for the C-terminal A2 tract (Ala123-Glu130). From *Figure 4c*,ii, we find that the motion is slowed by a factor two ($\tau_C^{apoN130} = 2.10 \pm 0.05ns$ and $\tau_C^{2.5:1rpL5:N130} = 4.56 \pm 0.06ns$) for the C-terminal residues and flexibility (*Figure 4c*,iii) is reduced by nearly half ($S^2_{apoN130} = 0.44 \pm 0.01$ and $S^2_{2.51rpL5:N130} = 0.76 \pm 0.01$). We also note that residues at the beginning of the A2 tract (Val119-Asp122), which were well fit when $\tau_c$ was fixed to the overall tumbling value, also exhibited restricted motion in the 2.5:1 rpL5:N130 state (by about 30%) as compared with apo N130. $\tau_{c,local}$ for Ser125 in N130 could not be determined because the resonance was overlapped in the apo state.

## Acknowledgements

We thank Dr. Steven Gygi for graciously sharing the NPM1 protein interaction list prior to deposition in BioGrid; Dr. Patrick Rodrigues and Mr. Robert Cassell for peptide synthesis; Dr. Victoria Frohlich, Dr. Jennifer Peters and Dr. Jamshid Temirov for training and assistance with light microscopy; Dr. Aaron Phillips for assistance with bioinformatics data analysis; Dr. Sivaraja Vaithiyalingam for assistance with ITC experiments; Dr. Moreno Lelli for assistance with 1 GHz NMR experiments and TGIR-RMN for access to the instrument; Mr. Patrick Fitzgerald and Dr. Douglas Green for the gift of eGFP plasmids and Dr. Douglas Green for providing access to fluorescence microscopy resources; Mr. Cheon-Gil Park for assistance with protein preparation; and Dr. Yuh Min Chook for the gift of N122 plasmid. Images were acquired at the Cell & Tissue Imaging Center which is supported by SJCRH and NCI P30 CA021765-35. Research conducted at ORNL's Spallation Neutron Source was sponsored by the Scientific User Facilities Division, Office of Basic Energy Sciences, US Department of Energy. Oak Ridge National Laboratory is managed by UT-Battelle, LLC under US DOE Contract No. DE-AC05-00OR22725. D.B. acknowledges support from the US National Institute of General Medical Sciences (F32GM113290). We thank Dr. Tanja Mittag and Dr. Charles Sherr for critical evaluation of the manuscript and for engaging discussions. Discussions with the members of the R. Kriwacki and T. Mittag labs were also appreciated.

## Additional information

### Funding

| Funder | Grant reference number | Author |
|---|---|---|
| National Institute of General Medical Sciences | 1R01GM115634 | Richard W Kriwacki |
| ALSAC | | Richard W Kriwacki |
| National Cancer Institute | P30CA21765 | Richard W Kriwacki |
| U.S. Department of Energy | DE-AC05-00OR22725 | Christopher B Stanley |
| Très Grandes Infrastructures de Recherche - Résonance Magnétique Nucléaire | TGIR-RMN-00625 | David Ban |
| National Institute of General Medical Sciences | 1R01GM083159 | Richard W Kriwacki |
| National Institute of General Medical Sciences | R01GM066833 | Ashok A Deniz |
| National Institutes of Health | R01CA082491 | Richard W Kriwacki |
| National Institute of General Medical Science | F32GM113290 | David Ban |

The funders had no role in study design, data collection and interpretation, or the decision to submit the work for publication.

### Author contributions

DMM, Conception and design, Acquisition of data, Analysis and interpretation of data, Drafting or revising the article; JAC, CSG, DB, PRB, CBS, AAD, Conception and design, Acquisition of data, Analysis and interpretation of data; AN, Acquisition of data, Analysis and interpretation of data; RWK, Conception and design, Analysis and interpretation of data, Drafting or revising the article

## Additional files

### Supplementary files

• Supplementary file 1. NPM1 binding proteins, their identified R-motifs and predicted NoLSs.

• Supplementary file 2. GO term associations of NPM1 binding proteins.

• Supplementary file 3. Multiple R-motifs identified within the human proteome.

### Major datasets

The following previously published datasets were used:

| Author(s) | Year | Dataset title | Dataset URL | Database, license, and accessibility information |
|---|---|---|---|---|
| Huttlin EL, Ting L, Bruckner RJ, Paulo JA, Gygi MP, Rad R, Kolippakkam D, Szpyt J, Zarraga G, Tam S, Gebreab F, Colby G, Pontano-Vaites L, Obar RA, Guarani-Pereira V, Harris T, Artavanis-Tsakonas S, Sowa ME, Harper JW, Gygi SP | 2016 | High-Throughput Proteomic Mapping of Human Interaction Networks via Affinity-Purification Mass Spectrometry (Pre-Publication) | http://thebiogrid.org/166968/publication/high-throughput-proteomic-mapping-of-human-interaction-networks-via-affinity-purification-mass-spectrometry.html | Publicly available at BioGRID (Accession no: Bait: NPM1). |

| | | | | |
|---|---|---|---|---|
| Uniport C | 2014 | UniProtKB - Q14978 (NOLC1_HUMAN) | http://www.uniprot.org/uniprot/Q14978 | Publicly available at UniProtKB (accession no. Q14978) |
| Uniport C | 2014 | UniProtKB - P19338 (NUCL_HUMAN) | http://www.uniprot.org/uniprot/P19338 | Publicly available at UniProtKB (accession no. P19338) |
| Scott MS, Boisvert FM, McDowall MD, Lamond AI, Barton GJ | 2010 | All human predicted NoLSs with Uniprot or RefSeq accessions | http://www.compbio.dundee.ac.uk/www-nod/downloads/AllPredicted-HumanNoLSs.txt | Publicly available at Nucleolar localization sequence Detector website (http://www.compbio.dundee.ac.uk/www-nod/downloads.jsp) |

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
