## [Decision Letter]

[Editors’ note: a previous version of this study was rejected after peer review, but the authors submitted for reconsideration. The first decision letter after peer review is shown below.]

Thank you for choosing to send your work entitled "NPM1 facilitates nucleolar assembly through phase separation with ribosomal components" for consideration at *eLife*. Your full submission has been evaluated by John Kuriyan (Senior editor) and three peer reviewers, one of whom served as a guest member of our Board of Reviewing Editors (Michael Rosen), and the decision was reached after discussions between the reviewers. Based on our discussions and the individual reviews below, we regret to inform you that your work will not be considered further for publication in *eLife*.

This manuscript reports on the in vitro properties of NPM1, an abundant nucleolar protein with many cellular functions. The authors use a suite of techniques, including fluorescence anisotropy, small angle neutron scattering, NMR spectroscopy, and FRAP to understand the molecular driving forces for assembly of the nucleolus. They find that the oligomerization domain of NPM1 can phase separate in the presence of peptides containing two R-rich motifs commonly found in NPM1 binding partners, including the ribosomal protein rpL5. They also demonstrate that full length NPM1 can phase separate with ribosomal RNA and the R motif-containing rpL5 peptide and uncover distinct roles for the oligomerization and nucleic acid-binding domains of NPM1 in droplet formation and morphology. Using NMR they show that the internal dynamics of NPM1 greatly slow within the phase separated droplets.

The referees all feel that these data are interesting, and provide new information regarding the mechanisms of protein phase separation. However, the in vitro data on NPM1 do not in themselves provide sufficient conceptual advance for publication in *eLife*. Such an advance would require strong connections to the cellular properties and behaviors of nucleoli, and here the paper falls short.

The authors wish to conclude from the data presented that "NPM1 facilitates nucleolar assembly through phase separation with ribosomal components" (title of paper). However there are no in vivo experiments to support this conclusion. This would require experiments that compare the effect of NPM1 mutants on droplet formation in vitro and nucleolar biogenesis in vivo, to demonstrate that the in vitro assay used here is relevant to nucleolar biogenesis.

In the last experiment of the paper, the author show that GFP:NPM1 accumulates in purified nucleoli and that NPM1 droplets formed in vitro associate with purified nucleoli. The authors offer these observations as evidence for in vivo relevance. However the pictures shown in Figure 12 indicate that the NPM1 droplets dock but do not fuse with the nucleoli, which does not support the authors assertion of "compatibility". Further, the specificity of the data in Figure 12 is not addressed. Using only DIC imaging to monitor the isolated nucleoli, it is unclear if the nucleoli are dissolving in the droplets or simply being surrounded by them. If the latter, is this behavior specific to nucleoli, or would the droplets adhere to and engulf any object they settled on? Without additional data linking the in vitro observations to an in vivo phenomenon, the main thesis of the paper (that NPM1 phase separation properties drive nucleolar biogenesis) is not supported.

Additional detailed comments on the manuscript and work are provided below.

1) The concentrations needed for phase separation are quite high. This raises questions about the contribution of Npm1 phase separation to nucleoli. However, this concern could be alleviated by further, quantitative studies of cooperativity between RNA and rpL5, akin to those in Figure 1. Cooperativity could drop phase separation into a more physiologic concentration range.

2) In general, for analytical measurements made above the phase separation concentration, the authors have not addressed how the heterogeneity of the solution will impact the data or interpretations thereof. This is an issue in the NMR data of Figure 5 and also the fluorescence anisotropy data of Figure 3.

2A) As far as I can tell, the NMR analyses of "the droplet state" were carried out on a mixture of droplet + bulk, rather than pure droplet state. Thus, the NMR behaviors reflect a complex average of the two states of the system, which will depend on the relative amounts of the two states (which is not provided) and their properties. While the qualitative features of the droplet state are likely correct (dynamics are slowed), this mixing calls into question the quantitative aspects of the analysis. The authors need to address this issue in their analyses and presentation of the work.

The fact that the relaxation data for the N- and C-termini could be fit globally does not in itself indicate that these elements undergo collective motions, as is stated in the text. If one assumes collective motion, then the global fitting can give greater precision in the derived values. But the interpretation does not go in the other direction in the absence of knowing what the errors should be if the motions were not collective. Phrased differently, what would the authors have observed in their analyses if the motions had been uncoupled but were fit as collective? Would the fitting have produced qualitatively different results? Or would the errors in the fitted parameters have simply been larger? If only the latter, at what point does one say the hypothesis of collective motion was wrong? Statistical analyses are needed to support this assertion.

2B) The fluorescence anisotropy experiments of Figure 3 will be complicated by light scattering by the droplets. The experiments are not described in sufficient detail to assess whether and how scattering was dealt with. Without this, it is hard to know how to interpret the break in the curve at ~x=2.5. Is this just the onset of scattering by the phase transition? Or does it represent a different tumbling rate of the molecules in the droplet phase?

3) The requirement for multivalency is not well-tested by the rpL5-RA peptide, as its affinity for N130 is appreciably lower than the other proteins examined. Thus, it is not clear whether the lack of phase separation results from the charged residues being clustered into a single block, or from the lower affinity. The data on the N122 protein make a stronger case (as does a comparison of the various R-motif proteins, where higher valency seems to correlate with a lower phase separation concentration). But the authors should seek a higher affinity mono-valent peptide in the first experiments.

4) The authors should also analyze their 2:1 SANS data in Figure 3, to see if higher-order species (beyond the assemblies seen a 1:1) begin to appear before phase separation occurs.

5) The molecular pictures shown in Figure 7 do not accurately reflect the authors' model. Comparing the 2:1 and 3:1 stoichiometries, it appears as though the fractional saturation of the A tracts decreases as more rpL5 is added. This cannot be the case. It is also not clear from the authors' description in paragraph four, Results, exactly what causes phase separation when the rpL5 concentration reaches the critical value. The authors describe the appearance of interpentamer crosslinks as occurring suddenly when rpL5 reaches some concentration and driving phase separation. But there should always be some such crosslinks. Affinity aside, one might expect that crosslinks would be most favored at low rpL5 stoichiometries. Maybe the key issue is solubility of the various species?

6) The data in Figure 11 are confusing, at least as described. What does the y-axis represent? "Fold-increase" over what fluorescence? The bulk (non-nucleolar) fluorescence at each point, or some initial fluorescence? If the former, so that each point represents the fold-increase of nucleolar fluorescence over background (bulk solution) fluorescence, then the data do not show saturation. Rather, in this case a constant fold-increase as the total gN294 rises means that the nucleolar fluorescence rises linearly. For a saturating system, the ratio of nucleolar/bulk fluorescence should decrease above the saturation point as the denominator grows and the numerator remains constant. If the y-axis is relative to some initial fluorescence value, the system does show saturation, but it is not clear what that initial point is.

7) I believe the squares and circles in Figure 4 should be reversed. The resonance in the core, currently indicated with the squares, should have faster transverse relaxation than those in the disordered A2 tract, currently indicated with the circles. But the opposite is observed in the data. The legend of Figure 4 should state more clearly what is shown in each panel. "Decay curve" doesn't say what the axes are.

8) In Figure 6, most of the change in FRET occurs upon binding rpL5 (e.g. in the 2.7:1 mixture in panel b), in the absence of phase separation. So the change is not dependent on phase separation, but rather rpL5 binding. This should be described differently in paragraph four, Results.

9) In the beginning of the Discussion, the authors should be more cautious in describing the physical nature of the droplets. Their model posits a "mesh-like lattice", but this is only indirectly supported by their data. In paragraph two, the authors should also be more cautious. It seems unlikely that NPM1 participates in every single chemical transaction that occurs within the GC.

10) There is a statement: "Thus, multivalent R-motifs are a hallmark of NPM1-bound nucleolar proteins". However, I think an enrichment from 44% of all human proteins to 73% of NPM-1binding proteins does not support the use of the word "Hallmark".

11) Paragraph three, Results: Explain FA technique briefly.

12) The discussion in the Results moves very quickly. It will be difficult for non-structural biologists (and probably even for structural biologists), such as would be the typical *eLife* reader, to follow. I suggest simplifying and using better transitions, highlighting the "take-home message" at each step etc.

13) In paragraph four, Results, it says that the second transition in the FA curve corresponds to phase separation. But this data seems to be taken at 200 µM N130. Looking at the phase diagrams in Figure 1, this is higher than all shown concentrations. How is the claim about the FA transition reflecting phase separation possible, since these concentrations do not appear to support phase separation?

14) I was confused by the statement in paragraph five, Results, that "the morphology of droplets" was strongly influenced by their composition. If they are liquids droplets in suspension, shouldn't they basically always have a spherical morphology? If they are true liquids but have some elongated shapes, it presumably reflects interactions with the surface (wetting) – this seems to be the case in the images in Figure 10. So is the statement about droplet morphology really about surface wetting behavior that may depend on composition?

15) Figure 10 – the title reads "…and accumulation in isolated nucleoli". But if I understand correctly there are no isolated nucleoli in this figure.

16) There are many abbreviations throughout the paper that make reading difficult.

---

## [Author Response]

[Editors’ note: the author responses to the first round of peer review follow.]

We regret that certain speculative aspects of the manuscript, which in retrospect should have been presented differently, were viewed as few major weaknesses by the three reviewers. We have revised the manuscript to avoid excessive speculation and to present the major conclusions more clearly. Furthermore, a weakness as regards biological relevance has been addressed through new cellular experiments and a novel bioinformatics analysis. These and additional new in vitro data have allowed us to address all of the reviewers’ concerns. With these improvements, we feel that our manuscript offers a significant conceptual advance in understanding the molecular mechanism of protein localization within the liquid-like matrix of the nucleolus.

The main points of the revised manuscript are as follows:

A principal goal was to understand the molecular mechanism of integration of NPM1 within the nucleolar matrix.We identify, through in vitro experiments, two types of multivalent features of NPM1 (oligomeric acidic tracts and nucleic acid binding domains) that mediate phase separation with two types of nucleolar components: i) proteins that display multivalent, arginine-rich short linear motifs (R-motifs) and ii) ribosomal RNA (rRNA).Importantly, the acidic low complexity regions, as well as the oligomerization-mediated multivalency are two novel classes of structural features associated with protein-protein and protein-RNA phase separation.Cellular experiments showed that both types of multivalent features are required for integration of NPM1 within the nucleolar matrix.These in vitro and cellular results suggest strongly that the ability of NPM1 to phase separate in vitro with nucleolar components is associated with its ability to integrate within the nucleolar matrix.The sequence features of proteins shown to cause phase separation with NPM1 in vitro match those of canonical nucleolar localization signals shown by others to mediate the nucleolar localization of hundreds of proteins.Therefore, we propose that the localization of a large proportion of nucleolar proteins is mediated by their ability to phase separate within the nucleolar matrix due to the three types of multivalent features identified in this study.

*[…] The referees all feel that these data are interesting, and provide new information regarding the mechanisms of protein phase separation. However, the in vitro data on NPM1 do not in themselves provide sufficient conceptual advance for publication in eLife. Such an advance would require strong connections to the cellular properties and behaviors of nucleoli, and here the paper falls short.*

In order to strengthen biological relevance and to emphasize a key conceptual advance, we re-focused the manuscript towards the identification of a currently unknown molecular mechanism responsible for protein targeting to nucleoli. To validate our in vitro findings, which show that both the ability to oligomerize and to bind nucleic acids is required for NPM1 to phase separate with ribosomal RNA and a multivalent R-motif-containing peptide, we created cell lines that stably express mCherry-NPM1 constructs that parallel those used in the in vitro studies. We show by fluorescence microscopy that only the full length protein can accumulate within mammalian nucleoli and co-localize with known nucleolar markers (Figure 12 and Figure 12—figure supplement 1). The major conceptual advance is that the protein sequence features (multivalent R-motifs) we have identified as being required for binding to and phase separation with NPM1 correspond to features identified by others as being nucleolar localization signals (NoLSs). We gained this insight by using bioinformatics tools to compare our list of NPM1 binding proteins with another from others of NoLSs in all human nucleolar proteins; we present this analysis in the Discussion of the revised manuscript. Thus, we propose that the molecular mechanism of protein phase separation with NPM1 that we describe is, in essence, the mechanism through which hundreds of proteins are localized within the nucleolus.

*The authors wish to conclude from the data presented that "NPM1 facilitates nucleolar assembly through phase separation with ribosomal components" (title of paper). However there are no in vivo experiments to support this conclusion. This would require experiments that compare the effect of NPM1 mutants on droplet formation in vitro and nucleolar biogenesis in vivo, to demonstrate that the in vitro assay used here is relevant to nucleolar biogenesis.*

We replaced the speculative title, “NPM1 facilitates nucleolar assembly through phase separation with ribosomal components,” with one that better reflects the specific findings of our study. The title now reads, “NPM1 integrates within the nucleolus via multi-modal interactions with proteins displaying R-rich linear motifs and rRNA”.

*In the last experiment of the paper, the author show that GFP:NPM1 accumulates in purified nucleoli and that NPM1 droplets formed in vitro associate with purified nucleoli. The authors offer these observations as evidence for in vivo relevance. However the pictures shown in Figure 12 indicate that the NPM1 droplets dock but do not fuse with the nucleoli, which does not support the authors assertion of "compatibility". Further, the specificity of the data in Figure 12 is not addressed. Using only DIC imaging to monitor the isolated nucleoli, it is unclear if the nucleoli are dissolving in the droplets or simply being surrounded by them. If the latter, is this behavior specific to nucleoli, or would the droplets adhere to and engulf any object they settled on? Without additional data linking the in vitro observations to an in vivo phenomenon, the main thesis of the paper (that NPM1 phase separation properties drive nucleolar biogenesis) is not supported.*

We agree with the reviewers that the conclusions from the “compatibility” experiments were speculative and therefore completely eliminated these data (original Figure 12) and the associated text from the revised manuscript. Rather, a more direct connection to in vivo relevance was added in the revised version of the manuscript in the form of cellular experiments, as discussed above.

*Additional detailed comments on the manuscript and work are provided below. 1) The concentrations needed for phase separation are quite high. This raises questions about the contribution of Npm1 phase separation to nucleoli. However, this concern could be alleviated by further, quantitative studies of cooperativity between RNA and rpL5, akin to those in Figure 1. Cooperativity could drop phase separation into a more physiologic concentration range.*

We agree with the reviewers that the concentration required for phase separation of N130 with the peptide is quite high. However, the critical concentration for the system drops ~5-fold when N130 is replaced with full length NPM1 (N294). This point is discussed in the current manuscript: “Furthermore, droplets of NPM1 with rpL5 formed at ~5-fold lower concentrations of both components (Figure 8) than were required for phase separation with N130 and rpL5 (Figure 1) due to the higher valency of acidic tracts within the full-length protein.” The data presented in Figure 8 and Figure 11 show that phase separation occurs in the tens of micromolar concentration regime with the divalent peptide, with ribosomal RNA and with both; we argue that this concentration regime is physiologically relevant.

*2) In general, for analytical measurements made above the phase separation concentration, the authors have not addressed how the heterogeneity of the solution will impact the data or interpretations thereof. This is an issue in the NMR data of Figure 5 and also the fluorescence anisotropy data of Figure 3. 2A) As far as I can tell, the NMR analyses of "the droplet state" were carried out on a mixture of droplet + bulk, rather than pure droplet state. Thus, the NMR behaviors reflect a complex average of the two states of the system, which will depend on the relative amounts of the two states (which is not provided) and their properties. While the qualitative features of the droplet state are likely correct (dynamics are slowed), this mixing calls into question the quantitative aspects of the analysis. The authors need to address this issue in their analyses and presentation of the work. The fact that the relaxation data for the N- and C-termini could be fit globally does not in itself indicate that these elements undergo collective motions, as is stated in the text. If one assumes collective motion, then the global fitting can give greater precision in the derived values. But the interpretation does not go in the other direction in the absence of knowing what the errors should be if the motions were not collective. Phrased differently, what would the authors have observed in their analyses if the motions had been uncoupled but were fit as collective? Would the fitting have produced qualitatively different results? Or would the errors in the fitted parameters have simply been larger? If only the latter, at what point does one say the hypothesis of collective motion was wrong? Statistical analyses are needed to support this assertion.*

We agree with the reviewers that we over-interpreted the NMR relaxation data for "the droplet state". In response to the reviewers’ comment, we performed new sedimentation velocity analytical ultracentrifugation experiments (Table 3 and Figure 3—figure supplement 1), which showed, indeed, that small N130 pentamer/rpL5 assemblies exist in equilibrium with the microscopically-detected droplets under conditions of phase separation. Therefore, we agree that we cannot interpret the NMR relaxation data for the droplet state as we did, and have eliminated these data from the manuscript. We do still briefly discuss the qualitative features of the 2D TROSY spectrum obtained under these conditions. Also, we still include the NMR relaxation analyses of N130 at N130:rpL5 stoichiometries below the threshold for phase separation, as follows:

“Above the phase separation threshold (>3:1 rpL5:N130), resonances for residues within the N130 core broadened beyond detection in 2D 1H-15N TROSY spectra but not those for residues within the A2 tract (Figure 4). Chemical shift values indicated that these residues remained disordered.”

*2B) The fluorescence anisotropy experiments of Figure 3 will be complicated by light scattering by the droplets. The experiments are not described in sufficient detail to assess whether and how scattering was dealt with. Without this, it is hard to know how to interpret the break in the curve at ~x=2.5. Is this just the onset of scattering by the phase transition? Or does it represent a different tumbling rate of the molecules in the droplet phase?*

This is a valid point raised by the reviewers. For the fluorescence anisotropy experiments, we used Alexa594 fluorophore labeled Npm-130^S125C^ (1 μM), mixed with unlabeled N130 (199 μM), with λ_ex_ = 580 nm, and λ_ex_ = 618 nm. Under these conditions, the light scattering contribution towards the fluorescence intensity due to liquid-liquid phase separation is < 5%, as measured using identical solution conditions lacking the labeled protein. Therefore, the 2nd transition in the anisotropy curve must represent the liquid-liquid phase separation with an altered tumbling rate of the N130 in the droplets, compared to the single-phase region in the phase diagram.

*3) The requirement for multivalency is not well-tested by the rpL5-RA peptide, as its affinity for N130 is appreciably lower than the other proteins examined. Thus, it is not clear whether the lack of phase separation results from the charged residues being clustered into a single block, or from the lower affinity. The data on the N122 protein make a stronger case (as does a comparison of the various R-motif proteins, where higher valency seems to correlate with a lower phase separation concentration). But the authors should seek a higher affinity mono-valent peptide in the first experiments.*

We rephrased the sentence to acknowledge the decrease in binding affinity:

“At the same N130 concentration, phase separation was not observed upon titration of a monovalent R-motif peptide (rpL5-RA; Figure 2), even though it bound, albeit with lower affinity (Table 2 and Figure 1—figure supplement 1), confirming that R-motif multivalency is required for phase separation with N130.”

Furthermore, in order to test the alternative explanation for the results proposed by the reviewer, namely that the lack of phase separation is caused by the weaker binding affinity, we performed light scattering assays of N130 titrations with a poly arginine peptide (RRRRRR), which has a similar binding affinity for the folded pentameric core of NPM1 as the divalent rpL5 peptide. In support of our model, this tighter binding, but monovalent, peptide did not phase separate with N130. The new data are presented in Figure 1—figure supplement 2 and the following phrase has been added to the Results section:

“The inability to phase separate was not due to reduced binding affinity (rather than loss of multivalency); a poly-R peptide, containing a single but longer R-motif, with affinity similar to that of rpL5, also failed to phase separate (Figure 2—figure supplement 1).”

*4) The authors should also analyze their 2:1 SANS data in Figure 3, to see if higher-order species (beyond the assemblies seen a 1:1) begin to appear before phase separation occurs.*

The analysis of the 2:1 SANS data is presented in the Analysis Methods section, in the subheading “SANS data analysis”. Additionally, we performed sedimentation velocity analytical ultracentrifugation assays to analyze the molecular weight distribution of the species present in the 0:1, 1:1, 2:1 and 3:1 rpL5:N130 samples (Table 3 and Figure 3—figure supplement 1). These additional results confirm the formation of higher order complexes at the 2:1 rpL5:N130 molar ratio, below the critical concentration at which microscopic droplets can be observed. We included a discussion of these results:

“Notably, nascent, higher order structural organization was evident in the SANS curve for a solution of rpL5 and N130 with 2:1 stoichiometry (Figure 3, green curve). […] The appearance of high molecular weight species within the detection range (< 1 MDa) was accompanied by a progressive decrease in the total mass detected, likely due to sedimentation of rpL5:N130 droplets in the sample cell during the dead-time of the experiment.”

*5) The molecular pictures shown in Figure 7 do not accurately reflect the authors' model. Comparing the 2:1 and 3:1 stoichiometries, it appears as though the fractional saturation of the A tracts decreases as more rpL5 is added. This cannot be the case. It is also not clear from the authors' description in paragraph four, Results, exactly what causes phase separation when the rpL5 concentration reaches the critical value. The authors describe the appearance of interpentamer crosslinks as occurring suddenly when rpL5 reaches some concentration and driving phase separation. But there should always be some such crosslinks. Affinity aside, one might expect that crosslinks would be most favored at low rpL5 stoichiometries. Maybe the key issue is solubility of the various species?*

We thank the reviewers for pointing out this source of confusion. We now explicitly state in the figure legend (Figure 6 in the revised manuscript) that “only a subset of the possible inter-N130 pentamer crosslinks are shown for clarity.” Additionally, we revised the text to better explain our model:

“As rpL5 is titrated into N130 up to 2:1 stoichiometry (rpL5:N130), the R1 motifs, comprised of four Arg residues each, bind to acidic residues within the A1 binding groove and disordered A2 tract of N130, with the R2 motif available for interactions. […]Together, our data support the hypothesis that the molecular basis of phase separation is the formation of non-covalent, inter-N130 pentamer interactions via the two R-motifs within the same rpL5 peptide molecule.”

*6) The data in Figure 11 are confusing, at least as described. What does the y-axis represent? "Fold-increase" over what fluorescence? The bulk (non-nucleolar) fluorescence at each point, or some initial fluorescence? If the former, so that each point represents the fold-increase of nucleolar fluorescence over background (bulk solution) fluorescence, then the data do not show saturation. Rather, in this case a constant fold-increase as the total gN294 rises means that the nucleolar fluorescence rises linearly. For a saturating system, the ratio of nucleolar/bulk fluorescence should decrease above the saturation point as the denominator grows and the numerator remains constant. If the y-axis is relative to some initial fluorescence value, the system does show saturation, but it is not clear what that initial point is.*

We replaced the Y axis labeling from “Fold-increase”to “I/I_background_ “in Figure 11 (Figure 12 in the revised manuscript) for clarity and modified the language in the legend correspondingly. As the reviewers correctly stated, the ratio of nucleolar vs. bulk fluorescence does decrease from ~ 25 at 20 μM eGFP-N294 to ~ 15 for 100 μM eGFP-N294 (compare Figure 12 with 12c), consistent with a model where the eGFP-N294 accumulation within isolated nucleoli is a saturable process. However, since we are not drawing any mechanistic conclusions based on this observation, we decided to not mention the saturation phenomenon in the revised manuscript.

*7) I believe the squares and circles in Figure 4 should be reversed. The resonance in the core, currently indicated with the squares, should have faster transverse relaxation than those in the disordered A2 tract, currently indicated with the circles. But the opposite is observed in the data. The legend of Figure 4 should state more clearly what is shown in each panel. "Decay curve" doesn't say what the axes are.*

We thank the reviewers for carefully examining the manuscript and pointing out this error. We corrected the erroneous legend. The Figure 4 legend now reads: “Circles and squares indicate data points for residues Tyr29 (within in the folded pentamer core) and Glu121 (within the disordered A2 tract), respectively.”

Additionally, we clarified the language in the legend for Figure 4, by specifying what the Y axis represents, as follows: “Examples of R_2,b_ and R_2,a_ peak intensity decay curves measured at 800 MHz (i) and 1000 MHz (ii), and R1 peak intensity decay curves measured at 800 MHz (iii) for free N130.”

8) In Figure 6, most of the change in FRET occurs upon binding rpL5 (e.g. in the 2.7:1 mixture in panel b), in the absence of phase separation. So the change is not dependent on phase separation, but rather rpL5 binding. This should be described differently in paragraph four, Results.

In smFRET measurements, the histograms reveal the identity of individual species in a heterogeneous mixture. In the single-phase solution, the native N130 pentamer is characterized by a single E_FRET_ peak ~ 0.85. When titrated against increasing [rpL5], there was no notable change in the histograms at a mole fraction of rpL5 (1:1 and 1:2) where binary complexes of N130 and the peptide were formed (as detected by FA, AUC and SANS), suggesting that the conformational features of N130 reported on by FRET were not altered significantly due to binding of the peptide in the single phase region of the phase diagram.

Droplet formation of N130 was initiated ~ 2.5 mole fraction (rpL5:N130). Therefore, at 2.7 mole fraction, we are measuring the conformational changes of N130 due to droplet formation or formation of higher order complexes, as indicated by AUC analysis, which was signaled by a relatively lower E_FRET_ peak. Under this condition, no liquid-like microscopic droplets can be observed. We also observe the native smFRET peak for N130 as well, which signifies the coexistence of both these states under this condition. The smFRET data > 3.0 mole fraction is essentially the same as the 2.7 mole fraction data, except that the native peak is significantly smaller, and total number of events are lower which may be due to less frequent diffusion of the larger droplets through the confocal volume. Also, sedimentation of the droplets can exclude the higher density phase from the detection volume, playing a role in lowering the number of events observed under this condition.

*9) In the beginning of the Discussion, the authors should be more cautious in describing the physical nature of the droplets. Their model posits a "mesh-like lattice", but this is only indirectly supported by their data. In paragraph two, the authors should also be more cautious. It seems unlikely that NPM1 participates in every single chemical transaction that occurs within the GC.*

We regret any confusion caused by use of the term "mesh-like lattice" in the original manuscript. We now avoid this term. Importantly, we provide additional evidence to support our model for the structural organization of N130/rpL5 liquid-like droplets. Specifically, we performed SANS experiments with droplets formed by N130 with a variant rpL5 peptide in which the linker between the two R-motifs was duplicated (rpL5-2xLinker; see Table 1). The results, shown in Figure 7, show that the longer peptide causes the correlation distances to become larger and to broaden. This suggests strongly that the rpL5 peptides, either wild-type or the 2xLinker variant, influence the inter-molecular spacing of N130 molecules within the liquid-like phase. We discuss these results in the revised manuscript, as follows.

“We propose that these rpL5-mediated interactions establish the inter-pentamer spacing within the droplet phase that was detected by SANS (Figure 3). […] We envision that such cross-links are dynamically formed and broken but that, once phase separation occurs, they dominate the structural organization detected by SANS.”

In addition, to elaborate on the “structural organization” of the N130/rpL5 liquid-like phase, we have added a new paragraph in the Discussion, as follows:

“In our structural studies of the minimal phase separating system, N130/rpL5, we showed that within the liquid-like matrix of the droplets, with diameters on the microns to tens of microns length scale, we observed local ordering on the length scale of 5 to 12 nanometers (Figure 3 and Figure 7). […] This model for thestructurally heterogenous integration of NPM1 within the nucleolar matrix heterogenous provides a mechanistic explanation for its central role into many nucleolar functions, such as ribosome biogenesis and nucleolar stress responses (Lindstrom, 2011).”

We hope that this clarifies our view of “structural organization” within the N130/rpL5 droplets and the relevance of this model to the much more complex and heterogeneous GC region of the nucleolus.

*10) There is a statement: "Thus, multivalent R-motifs are a hallmark of NPM1-bound nucleolar proteins". However, I think an enrichment from 44% of all human proteins to 73% of NPM-1binding proteins does not support the use of the word "Hallmark".*

We changed the language and eliminated the word “hallmark”. The sentence now reads: “Thus, multivalent R-motifs are enriched in proteins that bind to NPM1.”

*11) Paragraph three, Results: Explain FA technique briefly*.

Please see our response to point 8, above.

*12) The discussion in the Results moves very quickly. It will be difficult for non-structural biologists (and probably even for structural biologists), such as would be the typical eLife reader, to follow. I suggest simplifying and using better transitions, highlighting the "take-home message" at each step etc.*

Please see our response to point 16.

*13) In paragraph four, Results, it says that the second transition in the FA curve corresponds to phase separation. But this data seems to be taken at 200 µMN130. Looking at the phase diagrams in Figure 1, this is higher than all shown concentrations. How is the claim about the FA transition reflecting phase separation possible, since these concentrations do not appear to support phase separation?*

The phase diagrams in Figure 1 represent a sub-set of N130 concentrations at which phase separation was observed with all four multivalent R-motif containing peptides. rpL5 has the lowest valency of them all, and therefore has the highest concentration threshold for phase separation. Titrations of 200 μM N130 with rpL5 peptide and the phase separation threshold for this particular binary system are further described in Figure 2. We included the following statement, justifying the use of the 200 μM N130 concentration:

“At 200 μM N130, upon titration of the divalent rpL5 peptide, phase separation was observed when the rpL5:N130 ratio reached ~3:1 (Figure 2).”

*14) I was confused by the statement in paragraph five, Results, that "the morphology of droplets" was strongly influenced by their composition. If they are liquids droplets in suspension, shouldn't they basically always have a spherical morphology? If they are true liquids but have some elongated shapes, it presumably reflects interactions with the surface (wetting)* – *this seems to be the case in the images in Figure 10. So is the statement about droplet morphology really about surface wetting behavior that may depend on composition?*

The language in the new manuscript has been extensively revised. Furthermore, we reproduced the results using silane-coated slides, to avoid excessive interactions of the droplets with the surface. The slide treatment minimized artifacts arising from “wetting” effects for most of the conditions examined. We thus eliminated the “morphology of droplets” terminology from the revised manuscript and refer specifically to the size of the droplets as a parameter that varies between different types of droplets.

*15) Figure 10* – *the title reads "*…*and accumulation in isolated nucleoli". But if I understand correctly there are no isolated nucleoli in this figure.*

The figures have been extensively modified, so this point is no longer applicable in the revised manuscript.

16) There are many abbreviations throughout the paper that make reading difficult.

The language in the manuscript has been extensively revised to improve the flow and make reading less tedious. Additionally, we included a schematic representation of the different NPM1 constructs used in the revised manuscript in Figure 2.